# A Model of Errors in Transformers

**Suvrat Raju** [1]  and **Praneeth Netrapalli** [2]

## Abstract

We study the error rate of LLMs on tasks like arithmetic that require a deterministic output, and repetitive processing of tokens drawn from a small set of alternatives. We argue that incorrect predictions arise when small errors in the attention mechanism accumulate to cross a threshold, and use this insight to derive a quantitative two-parameter relationship between the accuracy and the complexity of the task. The two parameters vary with the prompt and the model; they can be interpreted in terms of an elementary noise rate, and the number of plausible erroneous tokens that can be predicted. Our analysis is inspired by an "effective field theory" perspective: the LLM's many raw parameters can be reorganized into just two parameters that govern the error rate. We perform extensive empirical tests, using Gemini 2.5 Flash, Gemini 2.5 Pro and DeepSeek R1, and find excellent agreement between the predicted and observed accuracy for a variety of tasks, although we also identify deviations in some cases. Our model provides an alternative to suggestions that errors made by LLMs on long repetitive tasks indicate the "collapse of reasoning" (Shojaee et al., 2025), or an inability to express "compositional" functions (Dziri et al., 2023). Finally, we show how to construct prompts to reduce the error rate.

## 1. Introduction

Large Language Models (LLMs) are versatile systems (Brown et al., 2020; Raffel et al., 2020) that have shown remarkable success on a variety of benchmarks (Laskar et al., 2023; Chang et al., 2024). However, state of the art models still make errors even on relatively simple problems (Huang et al., 2025). To understand and prevent these errors is an important problem.

With this broad motivation, we examine errors made by LLMs on a simple class of problems. We focus on a class of deterministic tasks in which the input, the output, or both comprise a sequence of tokens, each drawn from a small set, and study the variation of the error rate with the length of the task. This class includes decimal and binary arithmetic (Yuan et al., 2023; Shen et al., 2023; Maltoni & Ferrara, 2024; Zhang et al., 2024; Shrestha et al., 2025; Sun et al., 2025; Bertolazzi et al., 2025; Feng et al., 2025), some classic dynamic programming problems (Dziri et al., 2023), the tower of Hanoi (Shojaee et al., 2025), list reversal, and nested application of linear transformations. These problems are not of direct practical interest since LLMs can easily write code to solve them (Chen et al., 2021; Wang & Chen, 2023; Jiang et al., 2024; Yang et al., 2024). Nevertheless they provide a clean arena where LLM errors can be identified, modeled and ameliorated.

Our second objective is to demonstrate that LLMs can be treated like other natural systems, and techniques from the natural sciences can be utilized to study them quantitatively (Kaplan et al., 2020; Allen-Zhu & Li, 2024). We describe a quantitative model that predicts the accuracy of LLMs on this class of tasks, and then verify the model empirically.

Although LLMs have hundreds of billions of parameters, our final result has only two parameters that vary with the prompt and the specific model. These parameters have a simple interpretation — a small number, $r$ that can be related to an elementary "noise rate" per token, and an order-1 number, $q$, that describes the number of potential "error directions". We propose that the accuracy of the model, $a$, varies with the complexity of the task, $c$, as

$$a = \frac{1}{\Gamma(\frac{q}{2})}\gamma(\frac{q}{2}, \frac{q}{2rc^{2\alpha}}) \tag{1}$$

where $\gamma(x, y)$ denotes the incomplete gamma function

$$\gamma(x, y) = \int_0^y t^{x-1}e^{-t}dt, \tag{2}$$

$\Gamma(x) \equiv \gamma(x, \infty)$ is the gamma function, and we suggest fixing $\alpha = 1$.

We derive this formula in section 2, starting with the hypothesis that the operation of an LLM can be modeled via an

[1]International Centre for Theoretical Sciences, Tata Institute of Fundamental Research, Bengaluru, India [2]Google Deepmind, Bengaluru, India. Correspondence to: Suvrat Raju <suvrat@icts.res.in>.

*Proceedings of the 43rd International Conference on Machine Learning*, Seoul, South Korea. PMLR 306, 2026. Copyright 2026 by the author(s).

effective model that implements self-attention (Luong et al., 2015; Bahdanau et al., 2015; Vaswani et al., 2017). Errors arise because the parameters of the effective model differ slightly from the parameters required to make a correct prediction. These errors accumulate across the context and lead the output of each attention layer to differ from the correct output. If the projection of the final output vector onto an incorrect token becomes significant—which we assume happens when this error crosses a threshold—the model makes an incorrect prediction. A simple scaling argument, and additional assumptions about the probability distribution of the errors lead to formula (1).

Our formula provides a remarkably accurate characterization of errors made by LLMs in extensive empirical tests comprising 8 different tasks, 3 state-of-the-art LLMs, and 0.2 million distinct prompts. However, we find systematic deviations in some cases, and our formula fails completely in one example. We use this failure to deduce the presence of additional effects that are important in this case, and construct a modified problem where these effects are insignificant, so that the formula can be applied successfully.

The fact that a large number of parameters reorganize themselves into a small number of effective parameters—for the purpose of predicting errors—is reminiscent of the paradigm of "effective field theory" that is used in physics (Georgi, 1993; Burgess, 2020). For example, a fluid is fundamentally described by a large number of microscopic parameters. But, in simple experiments, its behaviour is captured by a small set of effective parameters such as its density and viscosity. This is analogous to what we see in our experiments.

We hasten to add that, in physics, the reorganization of fundamental parameters into effective parameters is understood via a mature mathematical framework (Polchinski, 1992) that we currently lack in the context of LLMs. Our investigation is only suggestive that a similar framework might be applied fruitfully to the study of large models.

### 1.1. Previous Work

The performance of LLMs on arithmetic has been studied extensively. For instance, it has been proposed that errors arise because they use "pattern matching" rather than algorithms (Nikankin et al., 2024) or due to incorrect tokenization (Singh & Strouse, 2024). Experiments with dynamic programming and multiplication led to the suggestion (Dziri et al., 2023) that LLMs lack the expressive power to compute compositional functions. The tower of Hanoi was studied in (Shojaee et al., 2025), where it was suggested that LLM reasoning collapses beyond a threshold.

We do find some evidence that LLMs adopt inconsistent algorithms for tasks like arithmetic. However, our broad perspective differs from previous investigations. We hypothesize that models make errors due to their failure to implement attention precisely in tasks that involve a large number of similar tokens. The agreement of the empirically observed error rate with our quantitative formula in several cases — including examples where algorithms are explicitly provided — provides evidence for this hypothesis.

This paper is organized as follows. In section 2, we describe our model for errors. We do not provide a rigorous analysis. Rather, we present a "physics-style" model, which involves several assumptions motivated by empirical observations but leads to a simple final formula. In section 3, we describe extensive empirical tests of this formula in several settings. In section 4, we provide one example of how our analysis can be used to ameliorate errors. We conclude with some comments on future work in section 5. The Appendices provide technical details.

**Conflict of Interest Disclosure.** In addition to Deepseek R1, Gemini Flash and Gemini Pro — which are developed by Google DeepMind — were used to empirically validate the theoretical formula for the error-rate of LLMs. PN is employed by Google DeepMind, and a Google Cloud Platform award to SR's home institution (ICTS-TIFR) was used to perform experiments on the Vertex AI platform.

## 2. Error Model

We study tasks where the LLM is prompted in natural language, with commands and examples accompanying the data necessary for the task. The model then generates a number of intermediate tokens (Wei et al., 2022) before producing a final output. We wish to abstract away from these complications, so as to arrive at a simple effective error model. We start by characterizing the class of tasks that we are considering; define an effective "complexity" parameter; and then turn to our error model.

We make a number of assumptions, with the $n^{\text{th}}$ assumption identified by the label [**An**]. Possible modifications of these assumptions are discussed in Appendix B.

### 2.1. Characterization of Tasks

At an abstract level, we are interested in the ability of the model to implement a deterministic mathematical function that maps a minimal sequence of input tokens $\mathcal{S}^{\text{in}}$ to a minimal sequence of output tokens $\mathcal{S}^{\text{out}}$. For example, in the case of addition of numbers of equal length $c$, we can imagine that this function takes, as input, a sequence

$$\mathcal{S}^{\text{in}} = [a_1, a_2, \ldots a_c, S, b_1, b_2, \ldots b_c, S], \qquad (3)$$

where $a_i$ and $b_i$ represent the digits of the two numbers and $S$ is a special token indicating the end of a number. We use

$l_{\text{in}} \equiv \text{len}(\mathcal{S}^{\text{in}})$. The expected output is a sequence

$$\mathcal{S}^{\text{out}} = [o_1, o_2, \ldots o_{c+1}, S], \tag{4}$$

comprising the digits of the sum. We use $l_{\text{out}} \equiv \text{len}(\mathcal{S}^{\text{out}})$. We are interested only in the accuracy of the map $\mathcal{S}^{\text{in}} \rightarrow \mathcal{S}^{\text{out}}$. This abstract formulation allows us to focus on this aspect, while relegating the other command and thinking tokens to the background.

Second, we study tasks where each token is drawn from a small set of possibilities—such as digits or bits. This implies that even when $l_{\text{in}}$ (or $l_{\text{out}}$) becomes large, the entropy of $\mathcal{S}^{\text{in}}$ (or $\mathcal{S}^{\text{out}}$) saturates instead of growing.

Third, the tasks we study can all be solved by an algorithm that operates repetitively but at, each step, pays attention to only a small number of tokens and disregards the rest of the context. For example, every step in the process of addition only requires us to attend to a digit each from both numbers and a carry.

### 2.2. Idealized Autoregressive Model

For each task, it is useful to imagine an *idealized* autoregressive model that solves the task with perfect accuracy. In particular, if $t_1, \ldots t_{l_{\text{in}}}$ represent the tokens for a valid input, $\mathcal{S}^{\text{in}}$, and $t_{l_{\text{in}}+1}, \ldots t_{l_{\text{in}}+l_{\text{out}}}$ represent the tokens for the desired $\mathcal{S}^{\text{out}}$, then we demand that

$$\mathcal{M}_{\text{id}}(t_1, \ldots t_n) = t_{n+1} \tag{5}$$

for $l_{\text{in}} \leq n < l_{\text{in}} + l_{\text{out}}$. Our notation suppresses intermediate tokens that the model might need to generate to process its input before producing an output token.

We imagine that the first layer in the idealized model is an embedding layer that maps a sequence of tokens into a sequence of vectors in a space of dimension $d_{\text{emb}}$.

$$\mathcal{E}_{\text{id}}(t_1, \ldots t_n) = (v_1, \ldots v_n) \tag{6}$$

These vectors are processed by a set of stacked layers, each of which comprises an attention layer followed by a nonlinear function. These stacked layers map a sequence of vectors in the embedding space to a single output vector.

$$\mathcal{L}_{\text{id}}(v_1, \ldots v_n) = w. \tag{7}$$

Finally, an output layer $\mathcal{O}(w) = t_{n+1}$ converts a vector back to the token that has the closest embedding vector. Therefore,

$$\mathcal{M}_{\text{id}} = \mathcal{O} \circ \mathcal{L}_{\text{id}} \circ \mathcal{E}, \tag{8}$$

where $\circ$ denotes composition in the obvious manner. The architecture is specified more explicitly in Appendix A.1.

The parameters of the idealized model are chosen once and for all so that the model gives the correct output on all possible inputs. Such a model exists (Merrill & Sabharwal, 2024; Yang & Chiang, 2024; Liu et al., 2024; Jiang et al., 2025; Merrill & Sabharwal, 2025; Li, 2025) provided we allow arbitrary precision and allow the model to generate an arbitrary number of chain of thought (CoT) tokens. It might be helpful to think of the idealized model as a Turing machine constructed using the attention mechanism (Pérez et al., 2019). Alternately, the idealized model can be thought of as a RASP-L program (Weiss et al., 2021; Zhou et al., 2024; Yang & Chiang, 2024) for the given task.

There might be multiple choices of parameters that lead to the correct output, since multiple Turing machines can perform a given task. One possible choice is to choose an idealized model where the total number of parameters is as small as possible. We leave further investigation of this ambiguity to future work.

### 2.3. Complexity Parameter

We denote the minimal number of tokens that must be processed by the idealized model, including CoT tokens, to be $c$. Our empirical tests involve tasks where we expect that this total number of tokens, $c$, will scale linearly either with $l_{\text{in}}$ or $l_{\text{out}}$.

The functional form (1) is invariant under rescaling of the complexity parameter. A rescaling, $c_{\text{new}} = \lambda c_{\text{old}}$, can be compensated by a redefinition $r_{\text{new}} = \lambda^{-2\alpha} r_{\text{old}}$. Therefore, we may either choose $c = l_{\text{in}}$ or $c = l_{\text{out}}$ for the examples in this paper. For tasks beyond those that we study here, a different choice of $c$ might be indicated.

### 2.4. Effective Model

When the LLM is given an appropriate prompt and an input sequence $\mathcal{S}^{\text{in}}$, it produces an output sequence of the form (4), which we denote by $\widetilde{\mathcal{S}}^{\text{out}}$ with length $\widetilde{l}^{\text{out}}$. If $\widetilde{\mathcal{S}}^{\text{out}} = \mathcal{S}^{\text{out}}$, the model's output is accurate and otherwise it is inaccurate. As we change $\mathcal{S}^{\text{in}}$, while keeping the rest of the prompt invariant, the operation of a given LLM defines a map between possible input sequences and an output: $\mathcal{S}^{\text{in}} \rightarrow \widetilde{\mathcal{S}}^{\text{out}}$. In principle, this map can be completely determined, at a fixed value of $c$, by prompting the LLM with all possible input sequences. (See Appendix A.2 for more discussion.)

We define an *effective model*, $\mathcal{M}_{\text{eff}}$ by the property that it autoregressively generates the tokens in $\widetilde{\mathcal{S}}^{\text{out}}$. This is precisely analogous to (5).

The effective model might depend sensitively on the prompt and the LLM being used. The virtue of introducing this construct is that it allows us to focus on the essential aspect of the model's input and output rather than examining the complicated details of the operation of the full LLM.

Our first assumption is that the effective model has the same

architecture as the idealized model but possibly with slightly different parameters.

$$\mathcal{M}_{\text{eff}} = \mathcal{O} \circ \mathcal{L}_{\text{eff}} \circ \mathcal{E}_{\text{eff}}, \tag{A1}$$

with the idealized embedding and stacked layers now replaced by effective maps

$$\mathcal{E}_{\text{eff}}(t_1, \ldots t_n) = (v_1, \ldots v_n); \; \mathcal{L}_{\text{eff}}(v_1, \ldots v_n) = w. \tag{9}$$

For simplicity we imagine that the output layer of the model is the same as above: it simply picks the token that is closest in embedding space.

## 2.5. Errors

The output produced by this effective model might differ from the output produced by the idealized model that makes no errors. We quantify this difference by an error vector before the final projection to an output token

$$\mathcal{L}_{\text{eff}} \circ \mathcal{E}_{\text{eff}}(t_1 \ldots t_n) - \mathcal{L}_{\text{id}} \circ \mathcal{E}_{\text{id}}(t_1 \ldots t_n) = \epsilon(\mathcal{S}^{\text{in}}, n) \tag{10}$$

Note that the error $\epsilon$ depends on the input sequence $\mathcal{S}^{\text{in}}$ and on the stage of the output $n$.

Our second assumption is that the effective model predicts an incorrect token when the length of this error vector crosses a threshold, $\tau$.

$$\mathcal{M}_{\text{eff}}(t_1 \ldots t_n) \neq \mathcal{M}_{\text{id}}(t_1 \ldots t_n) \quad \text{iff} \quad \epsilon(\mathcal{S}^{\text{in}}, n)^2 > \tau^2. \tag{A2}$$

This assumption is motivated by the structure of the assumed output layer. If the error vector is large enough in magnitude to generate overlap with some other token in the vocabulary, the probability of generating the correct output token becomes small.

Since the model needs to produce $l_{\text{out}}$ tokens, if one had naively assumed that the error vectors are independent and identically distributed for each output token, one would have been led to the conclusion that the accuracy — the probability of producing the entire output with no errors — would be given by $a = (1 - r)^{l_{\text{out}}}$ for some $r$.

Such a formula does not fit the empirical data well. Moreover, the analysis in Appendix A.3, suggests that the error vectors for different $n$ are correlated. To sidestep this issue of correlations, we proceed by rewriting the accuracy as

$$a = \text{Prob.}(\epsilon_{\max}(\mathcal{S}^{\text{in}})^2 < \tau^2), \tag{11}$$

where $\epsilon_{\max}(\mathcal{S}^{\text{in}})^2$ is the largest value of $\epsilon(\mathcal{S}^{\text{in}}, n)^2$ for $l_{\text{in}} \leq n < l_{\text{in}} + l_{\text{out}}$ with $\mathcal{S}^{\text{in}}$ fixed.

Next, we decompose

$$\epsilon_{\max}(\mathcal{S}^{\text{in}}) = \sum_{i=1}^{q} E^i(\mathcal{S}^{\text{in}}) w_i, \tag{12}$$

in terms of a basis of orthogonal unit vectors $w_i$ with real coefficients $E^i$ that depend on $\mathcal{S}^{\text{in}}$. The parameter $q$ counts the number of "effective directions" in which errors can plausibly be made. Although the vocabulary of the LLM is very large, the effective and idealized model function within a much smaller space of tokens and, even within that space, only a few tokens might compete with the correct token. Therefore, we expect $q$ to be an $\text{O}(1)$ number.

The error vector that enters (11) still depends on the input sequence but we are interested in the mean accuracy after averaging over all possible input configurations. We assume that the coefficients that enter the decomposition (12) are drawn from a Gaussian random distribution with mean $0$ and variance $v$.

$$E^i \sim \mathcal{N}(0, v). \tag{A3}$$

In words, we assume that the coefficients of the largest error vector produced during output generation, considered over all possible inputs, are distributed according to [A3]. Strictly speaking, this assumption makes sense when $c$ is large enough that the space of the input and output sequences $\mathcal{S}^{\text{in}}$ and $\mathcal{S}^{\text{out}}$ can be approximated by a continuous space, which is the case in the problems we study. The choice of a Gaussian is primarily motivated by simplicity as discussed in Appendix B.

Although we formally assume an isotropic variance for the error coefficients, it is only the sum of their squares that enters $\epsilon_{\max}^2$. Therefore, if they are larger along some dominant directions than other directions this can be captured by varying the value of $q$.

In Appendix A.3, we argue that the architecture of the model supports the assumption that the variance of the largest error scales with the length parameter as

$$v = \sigma^2 c^{2\alpha}, \tag{A4}$$

where $\sigma, \alpha$ are $c$-independent parameters.

The intuition is that the attention mechanism repetitively processes tokens through fixed query and key matrices. If these matrices differ from the correct matrices, this leads to erroneous attention coefficients for every vector in the context that can accumulate over the length of the context.

If these errors had been uncorrelated, we would have $\alpha = \frac{1}{2}$ in [A4]. However, when the vectors in the context are drawn from a limited set, these errors can be correlated leading the variance to scale quadratically rather than linearly. Empirically, we find that choosing $\alpha = 1$, leads to $\text{O}(1)$ values of $q$ that can be readily interpreted in terms of plausible error directions. The choice $\alpha = \frac{1}{2}$ provides comparably good fits but leads to larger values of $q$ that are harder to interpret. Therefore, we set $\alpha = 1$ in what follows. This choice is discussed further in Appendix A.3.

It is convenient to introduce $r = \frac{q\sigma^2}{\tau^2}$. Since we have not chosen a scale for the magnitude of the standard deviation or the threshold, it is only their ratio $\frac{\sigma}{\tau}$ that can be relevant. In terms of this variable, an evaluation of (11) leads us to our final formula (1) for the expected frequency with which the model accurately solves a given task. Formula (1) predicts that the accuracy maintains a plateau close to 1 for small values of $c$ but falls off as $c^{-2\alpha q}$ for large $c$. More details are provided in Appendix A.5.

## 3. Empirical Verification

We empirically test formula (1) across 8 tasks listed below on Gemini 2.5 Flash (Flash), Gemini 2.5 Pro (Pro) (Comanici et al., 2025) and DeepSeek R1 (DeepSeek) (Guo et al., 2025), all with thinking enabled. We provide brief descriptions of each task below. The precise prompts used in our experiments are listed in Appendix D.

The data displayed below was collected using a total of $2.0 \times 10^5$ distinct prompts; this significantly exceeds the amount of data used in previous empirical studies along these lines. By default, 200 samples were taken for each value of the complexity parameter, and separately for each model, although this number was occasionally larger or smaller. We have made this data public at zenodo.org/records/18281512. The error bars indicate 95% confidence intervals and are computed using the procedure outlined in Appendix C.

The best-fit values of $r$ and $q$ are provided in Appendix C.1. In each case, we find an $O(1)$ value of $q$, which provides confidence in our interpretation of $q$ as the number of "error directions".

### 3.1. List Reversal

In this task, the model is asked to output the elements of a given list in reverse order. The complexity parameter, $c$, is taken to be the length of the list. The variation of accuracy with $c$ is shown in Figure 1. The performance of Flash and DeepSeek is similar, whereas Pro shows markedly superior performance. This performance pattern is present in all the tasks below. We see that all three models obey (1) well, although the model overestimates the accuracy of Flash at small values of $c$.

### 3.2. Nested Linear Transformations

Given an initial value $C_0$, and lists $A, B$ of equal length, $c$, the model is tasked with iteratively calculating $C_{i+1} = A_i * C_i + B_i$. The lists are chosen so that all elements of $A, B, C$ are in the range [-9,9]. Figure 2 shows that all models obey (1) well.

It has been suggested (Dziri et al., 2023; Peng et al., 2024)

that LLMs cannot accurately perform tasks that involve composing elementary functions due to limitations in the expressive capacity of the transformer architecture. But modern LLMs utilize "chain of thought" (Wei et al., 2022) that provably increases the transformer's expressive power (Chen et al., 2024; Liu et al., 2024). Given the good fit between the observed accuracy and our model, on this classic compositional task, we suggest that the accumulation of noise in the attention mechanism might provide a better explanation for these errors.

### 3.3. Dynamic Programming

Given a list of length $c$, the model is tasked with finding the subsequence with the largest sum that involves no adjacent elements from the original sequence. This problem was previously studied in (Dziri et al., 2023). Although models are easily able to verbalize the algorithm to solve this task, we include a detailed pseudocode in the prompt. This is meant to rule out the possibility that the loss of accuracy with increasing $c$ is due to a collapse of reasoning (Shojaee et al., 2025). The results are shown in Figure 3 and we see that all models obey the expected curve.

### 3.4. Tower of Hanoi

We consider a slight embellishment of the tower of Hanoi task studied in (Shojaee et al., 2025). The model is asked to output the first $c$ moves required to move 10 disks from the first to the second tower. The moves must obey the standard rules — a larger disk can never be placed on top of a smaller disk. However, the disks are labelled with a random permutation rather than in ascending order of size to exclude memorized solutions. The model is given an explicit algorithm: at move $n$, it must move the disk corresponding to the least-significant 1 in the binary representation of $n$.

The simple algorithm provided in the prompt ensures the model does not have to "reason" to obtain the solution. Nevertheless, the empirical results shown in Figure 4, display the same collapse of accuracy for large $c$ that was noted in (Shojaee et al., 2025). The fit with our error model provides evidence that this is not due to a failure of reasoning but because of an accumulation of smaller random errors.

### 3.5. Vanilla Addition

The model is tasked with adding numbers of equal length, $c$. Figure 5 shows that the accuracy of Flash and DeepSeek obeys (1). However, the formula fails completely for Gemini Pro! This indicates a failure of one of our assumptions. We believe that [**A1**] fails. This assumption can only hold if the LLM's final answer can be replicated by an effective model whose parameters do not vary with $c$. It is possible the Pro chooses different algorithms for different lengths leading to

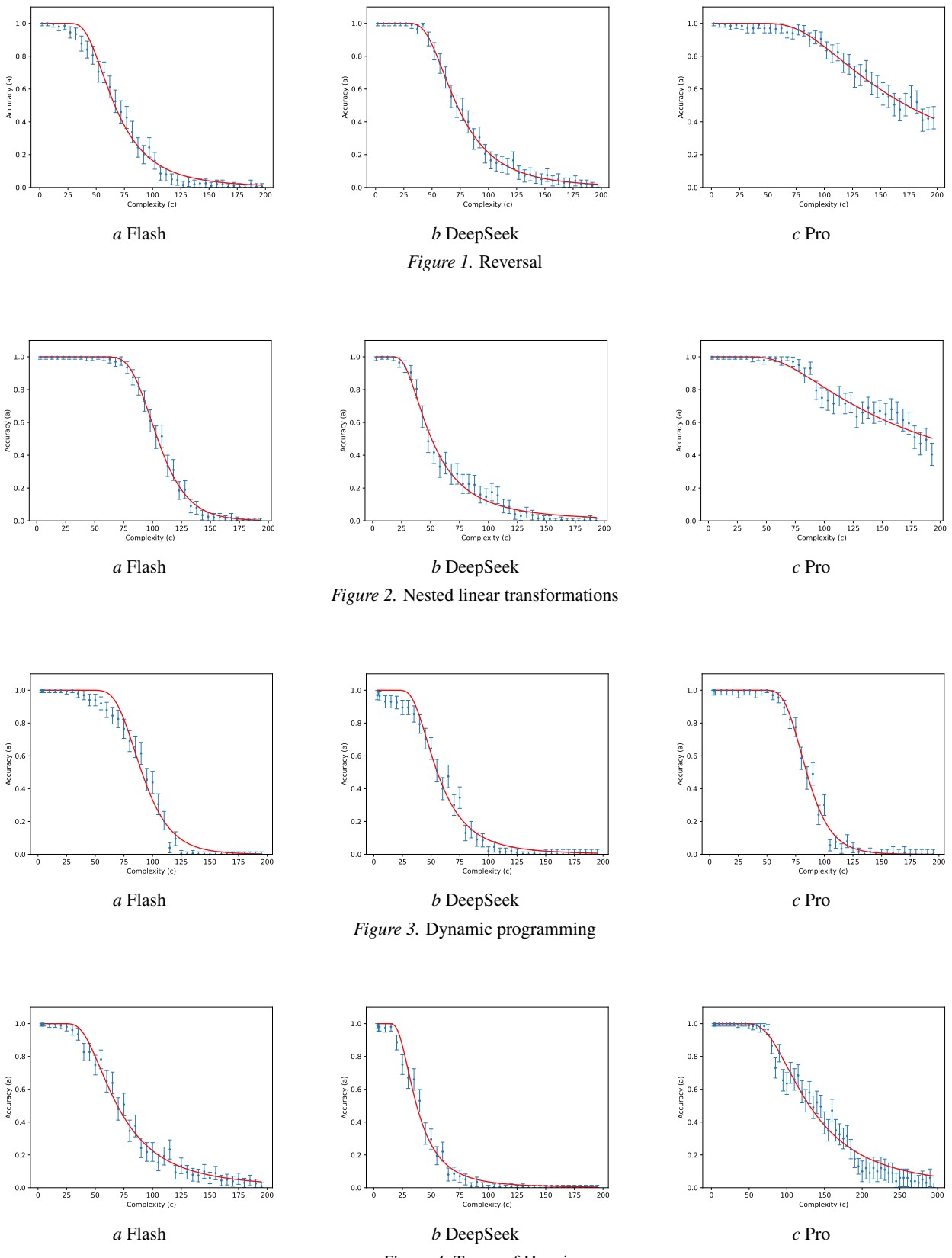

*a* Flash      *b* DeepSeek      *c* Pro

*Figure 1.* Reversal

*a* Flash      *b* DeepSeek      *c* Pro

*Figure 2.* Nested linear transformations

*a* Flash      *b* DeepSeek      *c* Pro

*Figure 3.* Dynamic programming

*a* Flash      *b* DeepSeek      *c* Pro

*Figure 4.* Tower of Hanoi

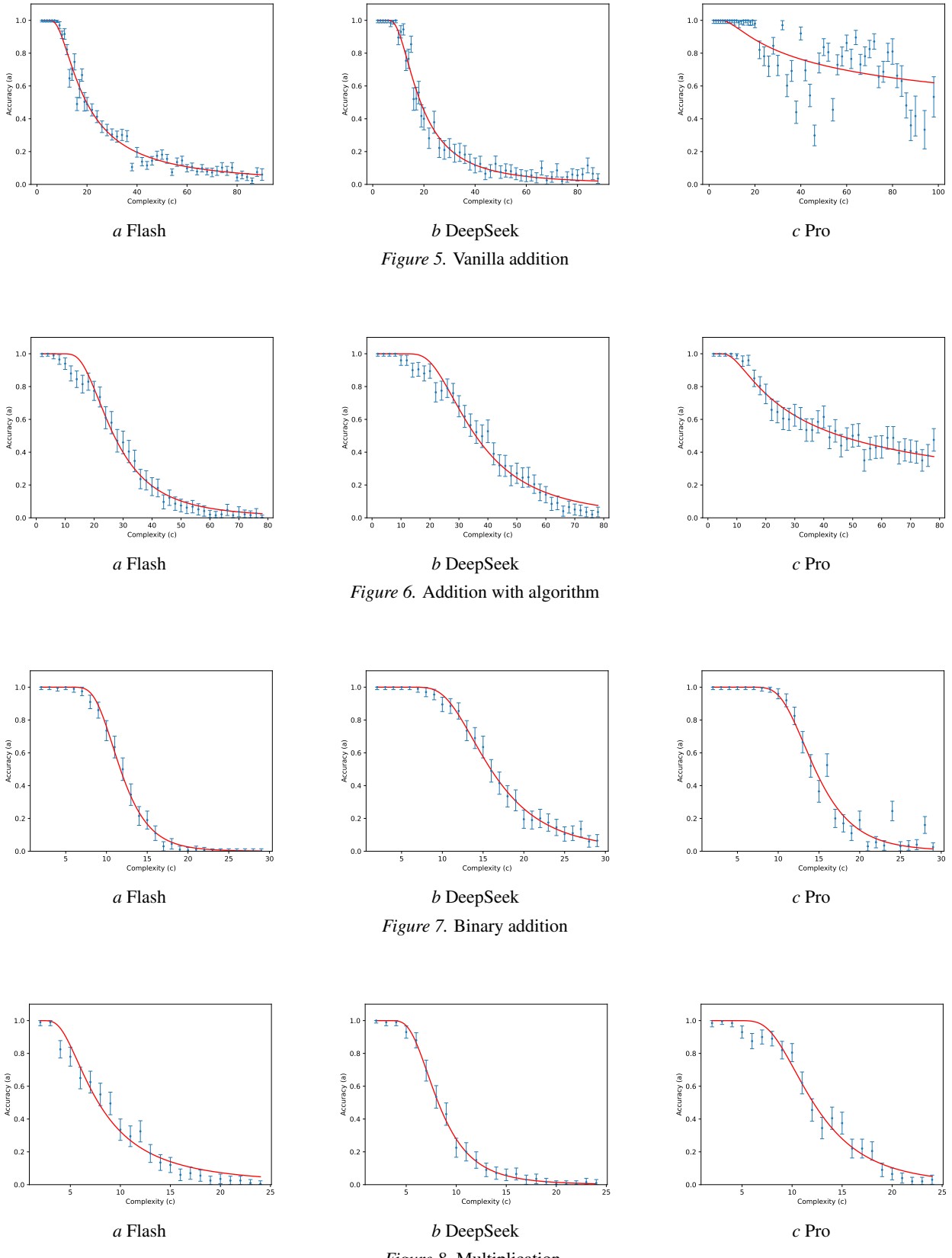

*a* Flash        *b* DeepSeek        *c* Pro

*Figure 5.* Vanilla addition

*a* Flash        *b* DeepSeek        *c* Pro

*Figure 6.* Addition with algorithm

*a* Flash        *b* DeepSeek        *c* Pro

*Figure 7.* Binary addition

*a* Flash        *b* DeepSeek        *c* Pro

*Figure 8.* Multiplication

an erratic pattern of errors.

### 3.6. Addition with Algorithm

To check the hypothesis of section 3.5, and better understand the failure of our formula for Pro, we revisit addition but force the model to obey a specific algorithm. Our prompt requires the model to decompose each number into a list of digits and then manipulate the digits to obtain a final sum. We eschew use of the word "addition" to discourage the model from using pre-learned algorithms. The model is asked to output the results of each intermediate step, although only the final sum is used to determine the accuracy. We now find in Figure 6 that all three models obey the curve (1). The fact that the empirically measured accuracy for Pro fits (1) when it is forced to use a specific algorithm bolsters the hypothesis of section 3.5.

### 3.7. Binary Addition

This task involves the addition of binary numbers of equal length, $c$, All models perform worse on this task than on decimal addition. Empirical results are shown in Figure 7, and all models obey (1) well.

### 3.8. Multiplication

The model is asked to multiply a fixed four-digit number, 7869, with numbers of varying length, $c$. This task was also studied in (Dziri et al., 2023). Figure 8 shows that all models obey (1) well. This provides additional evidence that errors arise due to the mechanism discussed in section 2 rather than a limitation in the LLM's expressive power.

## 4. Improving LLM Performance

Our experiments involved tasks where, at each step, the model's attention should have been focused on a small set of tokens. Nevertheless, the model appears to pay attention to irrelevant tokens in the context. Over a long context, these errors accumulate, leading to erroneous predictions.

This suggests that accuracy might be improved by tagging every token with an additional "tag token" that allows the model to focus its attention more sharply. To test this, we revisited the problem of multiplication using a modified prompt that instructs the model to translate numbers into polynomials, perform polynomial multiplication and then translate the result into a number. The token with information about the digit in the k$^{th}$ place is now tagged with a corresponding monomial $x^k$.

Figure 9 shows the accuracy of Flash with the modified prompt. This accuracy is superior not only to the accuracy of Flash but also that of Pro with a vanilla prompt.

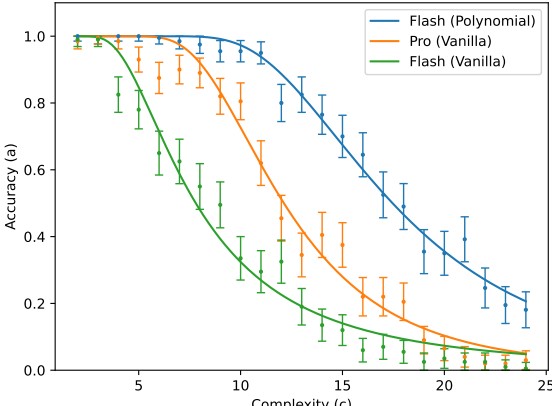

*Figure 9.* Accuracy of Flash and Pro on multiplication using different prompts.

## 5. Summary and Discussion

Our key structural assumption ([**A1**]) is that errors made by LLMs on a class of tasks can be analyzed through a smaller effective model, which differs slightly from an idealized model that solves the task with perfect accuracy. The form of the output layer suggests ([**A2**]) that an incorrect token is predicted when the difference in the vector predicted by the effective and idealized model crosses a threshold. Additional assumptions about the probability distribution of this error ([**A3**]), and scaling arguments for its dependence on the complexity ([**A4**]) lead to formula (1).

Our empirical tests span 8 different tasks and 3 different models with 0.2 million total test prompts. Formula (1) provides a surprisingly good description of the accuracy in almost all cases. In one case where the formula fails completely — vanilla addition with Pro — the failure suggests that the model might be using inconsistent algorithms for different input lengths. We provide evidence for this explanation by showing that the model's accuracy follows formula (1) when it is forced to follow a specific algorithm.

Modern LLMs are complicated systems, and although formula (1) performs well, it is unrealistic to expect a simple two-parameter fit to predict LLM accuracy perfectly. Indeed, our error model can be improved by replacing some of our assumptions with additional adjustable parameters. For instance, we ignored additive shifts of $c$ in the analysis of section (2). So, an obvious improvement is to introduce an additional parameter, $d$, and replace $c \to c+d$. Additionally, we could vary the parameter $\alpha$ that controls the scaling of the variance of the error. However, it is important to avoid overfitting the data while introducing additional parameters.

It is of interest to understand whether the techniques outlined in this paper can be generalized beyond the small class of tasks studied here, to understand errors in more-general

tasks that can be performed by LLMs.

In section 4 we showed that errors can be reduced by tailoring prompts that force the model to focus on relevant tokens in the context. More ambitiously, architectural improvements (DeepSeek-AI et al., 2025) might help models automatically weed out irrelevant tokens (Gu & Dao, 2024) in their context and reduce noise in the attention mechanism.

Our analysis suggests that models attain perfect accuracy on low-complexity tasks because they project onto a set of discrete tokens at each step. This serves as an error-correcting mechanism, and is an advantage of reasoning in token-space that should be balanced against possible benefits of reasoning in latent space (Hao et al., 2024).

## Acknowledgments

This research was supported by a GCP credit award to ICTS-TIFR. Research at ICTS-TIFR is supported by the Government of India through the Department of Atomic Energy via Project Identification No. RTI4001. We are grateful to members of the TIFR AI-ML consortium for helpful discussions and Jatin Batra and Karthikeyan Shanmugam for comments on a draft of this manuscript. Empirical data was collected using the Google VertexAI platform.

## Impact Statement

This paper presents work whose goal is to advance the field of machine learning. The empirical results presented in this paper required about 11K USD in compute resources with an associated environmental impact. There are other potential societal consequences of our work, none of which we feel must be specifically highlighted here.

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

# Appendix

# A. Technical Details for the Derivation of the Accuracy Formula

In this appendix, we provide additional technical details for the derivation presented in section 2.

## A.1. Architecture of the Idealized Model

The output of the idealized model is specified by (5). In words, the idealized model inputs a sequence of tokens and outputs one token with the property that its autoregressive operation transforms $\mathcal{S}^{\text{in}}$ to $\mathcal{S}^{\text{in}} + \mathcal{S}^{\text{out}}$ where $+$ stands for concatenation. As specified in section 2, it comprises an embedding layer, an attention layer and an output layer.

The vocabulary of the idealized model comprises a discrete set of possible tokens and is denoted by $\mathcal{T}$. In the cases studied in this paper, the vocabulary can be limited to digits and possibly a few special tokens for demarcating numbers.

### A.1.1. EMBEDDING LAYER

The embedding layer is a map

$$\mathcal{E}_{\text{id}} : \mathcal{T}^n \to \left(\mathbb{R}^{d_{\text{emb}}}\right)^n \tag{13}$$

that maps a sequence of input tokens to a sequence of vectors in a real $d_{\text{emb}}$-dimensional vector space (Mikolov et al., 2013). The simplest embedding is local in the tokens but our notation allows for a rich embedding that includes the incorporation of positional information (Vaswani et al., 2017; Su et al., 2024).

Positional information is essential to prove that attention-based models are Turing complete. Similarly, RASP programs assume that the model has information about the positional index of each token. We will assume that the embedding is rich enough for the idealized model to extract this information although we do not need to commit to a specific representation for our analysis.

Note that the value of $d_{\text{emb}}$ or the precise size of the vocabulary do not play a role in the final formula for the accuracy. This is an example of how when one restricts attention to a specific problem, the "fundamental parameters" of a model are replaced by "effective parameters."

### A.1.2. ATTENTION AND NONLINEAR LAYERS

The next layer of the idealized model is a map between a sequence of vectors to an output vector.

$$\mathcal{L}_{\text{id}} : \left(\mathbb{R}^{d_{\text{emb}}}\right)^n \to \mathbb{R}^{d_{\text{emb}}} \tag{14}$$

We imagine that this comprises a set of stacked layers, each of which applies self-attention to the input followed a nonlinear transformation. Given a sequence of input vectors, $v_i$, the first such layer maps it to a sequence of output vectors via

$$v_i^{(1)} = \phi(\mathcal{V}_i, v_i); \qquad \mathcal{V}_i = \sum_j A_{ij} v_j, \tag{15}$$

where $A_{ij}$ is an "attention matrix" that satisfies $\sum_j A_{ij} = 1$ and $\phi$ is a nonlinear map explained below. The attention matrix is itself computed as a function of the $v^j$ via

$$A_{ij} = \frac{\widetilde{A}_{ij}}{N_i} \tag{16}$$

with

$$\widetilde{A}_{ij} = e^{(Qv_i)\cdot(Kv_j)}; \qquad N_i = \sum_j \widetilde{A}_{ij}, \tag{17}$$

where $Q$ and $K$ are the "query" and "key" matrices respectively.

For transformer "decoders" the sum over $j$ that appears above is restricted to $j \leq i$, whereas it runs over all allowed values for transformer encoders. Since our simple analysis does not require us to commit to a choice, we leave the range implicit.

Some comments are in order.

1. In practice, it is convenient to break up the embedding space into different subspaces, and apply separate attention heads. For simplicity, our notation assumes that there is a single head where $Q, K$ are $d_{\text{emb}} \times d_{\text{emb}}$ matrices.

2. It is customary to divide the exponent in (17) by $\sqrt{d_{\text{emb}}}$. For the present analysis, this factor can be absorbed into $Q, K$.

3. We allow the idealized model to use arbitrary precision. When the operator norms of $Q, K$ are very large the softmax in (17) effectively becomes a hard-max that is commonly used in theoretical analyses of these models.

We have absorbed the "value" matrix, the layer norm (Ba et al., 2016) and the fully connected layer into a single nonlinear function $\phi$. Since $\phi$ depends both on $\mathcal{V}_i$ and $v_i$, it can also implement residual connections (He et al., 2016). Note that the action of $\phi$ does not depend on $i$.

Each subsequent layer is a map between sequences $v_i^{(q)} \to v_i^{(q+1)}$ using structurally similar functions, although the self-attention parameters $Q, K$ and the parameters that specify $\phi$ can differ between the layers. If there are $d$ layers, the output of $\mathcal{L}_{\text{id}}$ is simply $v_n^{(d)}$.

We have consciously not been more specific about the precise number of attention layers in $\mathcal{L}_{\text{id}}$, or the precise form of $\phi$. It is possible to obtain a Turing-complete architecture (using arbitrary precision) with only a few layers and nonlinear functions $\phi$ comprising feed-forward networks. Similarly, in the RASP-L framework every attention layer can be followed by an arbitrary token-level manipulation that is captured by $\phi$. Since our current analysis is very simple, we do not need to commit to these choices although they might be relevant for more-detailed work.

We are also not concerned with whether the weights of the idealized or effective model can be obtained by training. Our objective here is not to train the model but find a reference with which one can compare a simplified representation of the functioning of a trained LLM.

### A.1.3. CHAIN OF THOUGHT TOKENS

The architecture above defines a map between a sequence of input vectors and a sequence of output vectors. However, it does not provide a mutable memory that is generally required by a Turing machine. This obstacle can be overcome by generating intermediate tokens that are processed autoregressively prior to generating an output token. These tokens can be thought of as CoT tokens. We refer the reader to (Pérez et al., 2019) or (Zhou et al., 2024) for more discussion of these tokens. We would like to make a few comments.

1. Although our notation suppresses these tokens and focuses only on the final output tokens (see, for example, (5)) it should be kept in mind that they are necessary for most tasks.

2. The intermediate tokens being discussed here are those produced by a simple idealized model that solves a specific task. They should be distinguished from the CoT tokens produced by the operation of the full LLM.

3. The complexity parameter, $c$, defined in section 2.3 includes these tokens. The number of intermediate tokens produced by the idealized model can be estimated in terms of the number of steps required by a Turing machine to solve the task. For the simple tasks considered here, we expect that the number of such tokens will scale linearly either with $l_{\text{in}}$ or with $l_{\text{out}}$. (For most of our tasks, $l_{\text{in}}$ and $l_{\text{out}}$ scale together, although for the Tower of Hanoi, $l_{\text{in}}$ remains fixed and it is only $l_{\text{out}}$ that is varied.) Given this, we utilize the fact that the form of Formula 1 is invariant under the rescaling $c \to \lambda c$, to set $c = l_{\text{in}}$ or $c = l_{\text{out}}$.

### A.2. Effective Model

The effective model is defined as the map that a particular LLM induces between the input sequence and the output sequence upon being given a prompt.

An example might help to clarify this notion. Consider the prompt for "vanilla addition" given in section D.5. The two numbers that appear in this prompt specify the input sequence $\mathcal{S}^{\text{in}}$. The prompt itself is a longer sequence of tokens that can be denoted as $p(\mathcal{S}^{\text{in}})$. On being given $p(\mathcal{S}^{\text{in}})$ as an input, the LLM produces a sequence of output tokens where the subsequence starting after 'ANSWER:' can be identified as $\widetilde{\mathcal{S}^{\text{out}}}$. We define $\widetilde{l}^{\text{out}} = \text{len}(\widetilde{\mathcal{S}^{\text{out}}})$.

For any given value of $l_{\text{in}}$, the possible number of input sequences are finite. In principle, we can run the LLM on all possible prompts, keeping *all parts* of the input fixed except for the data $\mathcal{S}^{\text{in}}$. This defines a map between $\mathcal{S}^{\text{in}}$ and an output sequence, $\widetilde{\mathcal{S}}^{\text{out}}$, and thereby completely specifies $\mathcal{M}_{\text{eff}}$.

If $t_1, \ldots t_{l_{\text{in}}}$ and $t_{l_{\text{in}}+1}, \ldots t_{l_{\text{in}}+\widetilde{l}^{\text{out}}}$ specify the tokens that make up $\mathcal{S}^{\text{in}}$ and $\widetilde{\mathcal{S}}^{\text{out}}$ respectively then we demand that

$$\mathcal{M}_{\text{eff}}(t_1, \ldots t_n) = t_{n+1}. \tag{18}$$

for $l_{\text{in}} \leq n < l_{\text{in}} + \widetilde{l}^{\text{out}}$. The difference between the equation above and (5) is that this model generates the elements of $\widetilde{\mathcal{S}}^{\text{out}}$ rather than $\mathcal{S}^{\text{out}}$.

If the idealized model produces intermediate tokens prior to producing the output then so will the effective model. This follows from our demand that the effective model have the same architecture as the idealized model. The constraint (18) only applies to the output tokens and not to these intermediate tokens.

A few cautionary points are in order

1. When the temperature is nonzero, the output of the LLM has a probabilistic aspect. However, the analysis in the text only depends on the output vector produced by the model rather than the output token. The output vector is deterministic even in the presence of a nonzero temperature. Therefore, the temperature does not play any role in our analysis.

2. The LLM might not produce a valid output at all for some possible input sequences. In the example above, its output might not contain the keyword 'ANSWER:'. In such cases, we can set $\widetilde{\mathcal{S}}^{\text{out}}$ to be the null sequence or some other default sequence. The analysis in the text assumes that for most $\mathcal{S}^{\text{in}}$, $\widetilde{\mathcal{S}}^{\text{out}}$ is "close" to $\mathcal{S}^{\text{out}}$. Therefore, it is only applicable if the number of cases where the LLM's output is malformed are small in number. Empirically, this assumption is valid for state-of-the-art LLMs, which follow instructions accurately.

### A.3. Scaling of the Variance

To complete our argument, we need to justify [**A4**] and also the choice $\alpha = 1$, which posits that the variance of the largest error vector scales quadratically with the complexity parameter.

We start with [**A1**], where we assumed that the effective model differs from the idealized model via small errors in the parameters of the attention matrix and in the other functions. Consider the first layer of $\mathcal{L}_{\text{eff}}$. If this function is given the same inputs as the first layer of $\mathcal{L}_{\text{id}}$ then, after the application of attention, it obtains the outputs $\mathcal{V}_i + \delta\mathcal{V}_i$ with the error

$$
\begin{aligned}
\delta\mathcal{V}_i &= \frac{1}{N_i} \sum_j (\delta\widetilde{A}_{ij}) v_j - \frac{\delta N_i}{N_i^2} \sum_j \widetilde{A}_{ij} v_j \\
&= \frac{1}{N_i} \sum_j (\delta\widetilde{A}_{ij}) v_j - \Big(\sum_j \delta\widetilde{A}_{ij}\Big) \frac{\mathcal{V}_i}{N_i}
\end{aligned} \tag{19}
$$

The errors $\delta\widetilde{A}_{ij}$ can be further related to errors in the fundamental parameters, $Q + \delta Q$ and $K + \delta K$ of the effective model via (17).

$$\delta\widetilde{A}_{ij} = \widetilde{A}_{ij}\left((\delta Q v_i) \cdot (K v_j) + (Q v_i) \cdot (\delta K v_j)\right), \tag{20}$$

Recall that in the tasks under consideration, we expect the model to pay attention to only a few tokens, which means that at each step in token generation, most tokens in the context are irrelevant. This might naively have suggested that even if the model is acting on an input of length $c$, the effective size of the context at each step is much smaller.

However, our empirical observations suggest that errors occur precisely because the model pays attention to irrelevant tokens in the context. Recall that our interest is in the largest error vector produced during output generation. In the worst case, the effective context of the model scales linearly with the length parameter, $c$, which implies that the largest error vector on the left hand side of (19) can involve a sum over $\text{O}(c)$ vectors. So the expected magnitude-squared of this vector is

$$\max_i |\delta V_i|^2 \propto c^{2\alpha}. \tag{21}$$

If the error vectors that appear in (19) are uncorrelated, we expect the variance of the sum to scale linearly with $c$ i.e. $\alpha = \frac{1}{2}$. However, if these errors are correlated, we expect $\alpha = 1$. We return to the value of $\alpha$ in subsection A.4 below.

The addition of a nonlinear layer and additional stacks of attention layers does not change the scaling of the error with $c$. If the effective model implements a slightly incorrect nonlinear function $\phi + \delta\phi$, then the output after the application of that function is simply $v_i^{(1)} + \delta v_i^{(1)}$ with the error

$$\delta v_i^{(1)} = \delta\phi(\mathcal{V}_i, v_i) + \frac{\partial\phi(\mathcal{V}_i, v_i)}{\partial\mathcal{V}_i}\delta\mathcal{V}_i \tag{22}$$

Since the magnitude of $\delta\phi$ is independent of context length, and it is only the magnitude of $\delta\mathcal{V}_i$ that scales with $c$, we see that, in the limit of large $c$

$$\max_i |\delta v_i^{(1)}|^2 \propto c^{2\alpha}. \tag{23}$$

The argument proceeds inductively for additional stacks of attention and fully connected layers. It is easy to see that while the variance of the error increases with the depth, the scaling with $c$ is unaffected. We have assumed that the architecture of the effective model is fixed once and for all and its depth does not scale with the context. Therefore, the scaling of the error vector produced in the last layer for the last token also obeys (21).

The autoregressive structure of the model also explains why errors at each step are correlated with errors in the previous step. The attention parameters are fixed during the entire output generation. So, if the model has computed an incorrect key for tokens early in the context, this incorrect key continues to contribute to errors throughout the output generation. This is why it makes sense to analyze the largest error vector encountered during generation rather than imagining independent error probabilities for each output token.

In this paper, we have been focused on the absolute accuracy of the model — where the output is considered accurate when it does not contain a single mistake. For this notion of accuracy, (11) is exact and our worst-case analysis for the scaling of the variance is appropriate. One could consider other notions of accuracy — such as the number of errors normalized by $c$ — which would involve a more complicated analysis.

### A.4. Value of $\alpha$

We now turn to the value of $\alpha$. One possibility that can induce correlations between the terms of (19) is as follows. We have envisioned an embedding layer that adds positional information to the input. However, if the attention layer does not appropriately use this positional information, then the vectors that appear in (19) are themselves effectively drawn from a small discrete set with a size no larger than the number of tokens. But this means that for large $c$ these errors will add constructively, leading to $\alpha = 1$. On the other hand, uncorrelated errors would lead to $\alpha = \frac{1}{2}$. We start by discussing these two intuitive choices for $\alpha$, and then turn to more general values.

Empirically, we found both $\alpha = 1$ and $\alpha = \frac{1}{2}$ give comparably good fits to the data. However, the choice $\alpha = 1$ consistently yields O$(1)$ values of $q$ that can be interpreted in terms of a small number of plausible erroneous tokens that can be produced at each step. In contrast, we obtain larger values of $q$ with the choice $\alpha = \frac{1}{2}$.

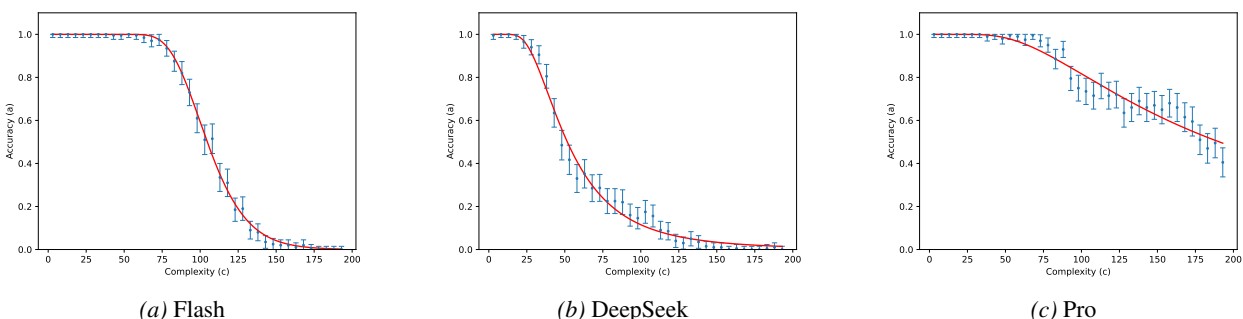

*(a)* Flash        *(b)* DeepSeek        *(c)* Pro

*Figure 10.* Nested linear transformations. The data points are the same as Fig. 2 but the solid line shows a fit with $\alpha = \frac{1}{2}$ instead of $\alpha = 1$.

An illustrative example is provided by the empirical data from the "nested linear transformations" task described in section

3.2. In Figure 10 we show the fit obtained with $\alpha = \frac{1}{2}$. This should be compared with Figure 2.

At a visual level, the fits are similar. However, a more detailed check can be performed by using the $\chi^2$ measure of the goodness of fit described in Appendix C. The $\chi^2$ and $q, r$ values for the two choices of $\alpha$ are shown in Table 1

| | Gemini Flash | | | DeepSeek | | | Gemini Pro | | |
|---|---|---|---|---|---|---|---|---|---|
| $\alpha$ | $r$ | $q$ | $\chi^2$ | $r$ | $q$ | $\chi^2$ | $r$ | $q$ | $\chi^2$ |
| $\frac{1}{2}$ | $(9.68 \pm 0.03) \times 10^{-3}$ | $53 \pm 2$ | 0.06 | $0.0210 \pm 0.0002$ | $8.6 \pm 0.4$ | 0.4 | $(6.36 \pm 0.07) \times 10^{-3}$ | $3.7 \pm 0.3$ | 0.3 |
| $1$ | $(9.64 \pm 0.07) \times 10^{-5}$ | $13.9 \pm 0.6$ | 0.09 | $(5.0 \pm 0.1) \times 10^{-4}$ | $2.8 \pm 0.1$ | 0.7 | $(5.7 \pm 0.2) \times 10^{-5}$ | $1.01 \pm 0.07$ | 0.2 |

*Table 1.* Comparison of parameters and goodness-of-fit for $\alpha = \frac{1}{2}$ and $\alpha = 1$ on the nested linear transformations task

The $\chi^2$ values show that the fits are similar, although $\alpha = \frac{1}{2}$ fits the data for Flash and DeepSeek slightly better and $\alpha = 1$ fits the data for Pro slightly better. In this task, each predicted number in the true answer is between $-9$ to $9$, which would suggest that there are, at most, 19 possible reasonable predictions that can be made at each step. The $q$ values with $\alpha = 1$ can therefore be interpreted in terms of plausible error directions. But the $q$ values with $\alpha = \frac{1}{2}$ have a much larger range and no such clear interpretation. This leads us to prefer the value $\alpha = 1$.

In addition to the two choices discussed here, an alternative is to fit $\alpha$ empirically in each instance. We did not do so to avoid increasing the number of parameters without a clear improvement in the empirical fit.

A quick assessment of the performance of other values of $\alpha$ is provided by averaging the value of $\chi^2$ for a fixed $\alpha$ across all models and all tasks of section 3. We denote this average by $\chi^2_{\text{mean}}$ and find the curve displayed in Figure 11. The numerical

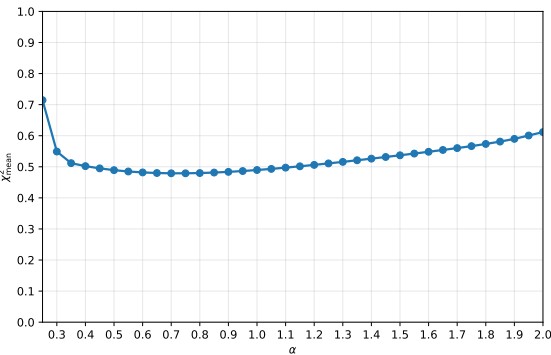

*Figure 11.* $\chi^2_{\text{mean}}$ obtained by averaging $\chi^2$ across all tasks and models for different choices of $\alpha$

values of $\chi^2_{\text{mean}}$ for selected value of $\alpha$ are displayed below.

| $\alpha$ | 0.25 | 0.50 | 1.00 | 2.00 |
|---|---|---|---|---|
| $\chi^2_{\text{mean}}$ | 0.71 | 0.49 | 0.49 | 0.61 |

As expected the goodness-of-fit for $\alpha = 0.5$ and $\alpha = 1$ is comparable, whereas choosing $\alpha < 0.5$ or $\alpha > 1.0$ yields a poorer fit. Numerically, we find that the minimum value of $\chi^2_{\text{mean}}$ is

$$\chi^2_{\text{mean}}\big|_{\alpha=0.74} = 0.48. \tag{24}$$

This value suggests that "on average", our experiments are described by an intermediate regime, with partially correlated and partially uncorrelated errors. Of course, the precise value of $\alpha$ at which the minimum is attained is specific to the combination of models and tasks that we are considering.

While the goodness-of-fit does not change much as $\alpha$ is varied between $0.5$ and $1.0$, the values of $q, r$ are more sensitive. This is displayed in Figure 12. Note that we restrict $q \leq 100$, although this cutoff is only relevant at small values of $\alpha$ for a limited combination of tasks and models.

This analysis supports the assertion made in the paper that the choice $\alpha = 1$ yields an easily interpretable formula, with no significant loss of goodness-of-fit.

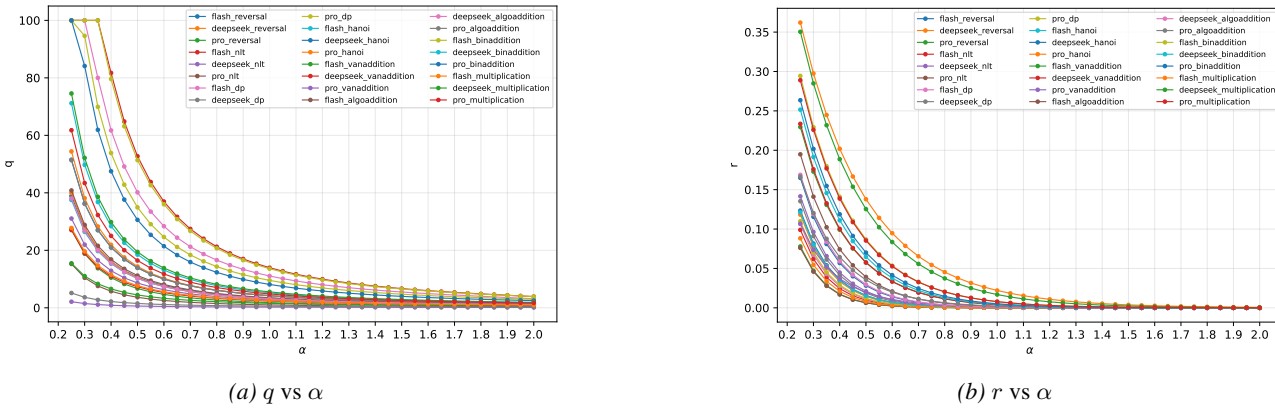

*(a) q vs α*                                          *(b) r vs α*

*Figure 12.* Variation of fit parameters with $\alpha$ for all combinations of models and tasks.

## A.5. Calculation of Accuracy

Finally, we present the short calculation leading to (1). The assumption [**A3**] leads us to

$$P(\epsilon_{\max}^2 < \tau^2) = \int \theta(\tau^2 - \sum_i E_i^2)\left(\frac{1}{\sqrt{2\pi v}}\right)^q e^{-\sum_i \frac{E_i^2}{2v}} \prod_i dE_i \tag{25}$$

Changing to polar coordinates with $t = \frac{1}{2v}\sum_i E_i^2$, we see that

$$P(\epsilon_{\max}^2 < \tau^2) = \frac{1}{\Gamma(\frac{q}{2})} \int_0^{\frac{\tau^2}{2v}} e^{-t} t^{\frac{q}{2}-1} dt \tag{26}$$

where the factor of $\frac{1}{\Gamma(\frac{q}{2})}$ arises by combining the volume of the $(q-1)$-sphere obtained by transforming to polar coordinates with the normalization of the Gaussians. However, it can also be seen to be correct since the probability must tend to 1 when $\tau^2 \to \infty$.

Using the definition of the lower incomplete gamma functions (DLMF), denoted below by $\gamma$, we immediately find that

$$P(\epsilon_{\max}^2 < \tau^2) = \frac{\gamma(\frac{q}{2}, \frac{\tau^2}{2v})}{\Gamma(\frac{q}{2})}. \tag{27}$$

After substituting the definition of $r$ and [**A4**] so that $\frac{\tau^2}{2v} = \frac{q}{2rc^{2\alpha}}$ we arrive at the final formula for the accuracy

$$a = P(\epsilon_{\max}^2 < \tau^2) = \frac{1}{\Gamma(\frac{q}{2})}\gamma(\frac{q}{2}, \frac{q}{2rc^{2\alpha}}), \tag{28}$$

as advertised. Note that a factor of $q$ is explicitly inserted into the normalization for $r$, which is why $q$ appears in the second argument in the numerator.

Note that SciPy (Virtanen et al., 2020) uses a different normalization for the incomplete gamma function, and the ratio of the incomplete gamma function and gamma function that appears in (28) is implemented as `scipy.special.gammainc`.

For large $c$, the formula (28) can be expanded as

$$a \xrightarrow[c\to\infty]{} \frac{1}{\Gamma\left(\frac{q}{2}+1\right)}\left(\frac{q}{2rc^{2\alpha}}\right)^{q/2}, \tag{29}$$

which shows a power-law decay of the accuracy in this regime. However, at small $c$, we find that

$$a \xrightarrow[c\to 0]{} 1 - \frac{\left(\frac{q}{2rc^{2\alpha}}\right)^{\frac{q}{2}-1} e^{-\frac{q}{2rc^{2\alpha}}}}{\Gamma\left(\frac{q}{2}\right)}, \tag{30}$$

which shows that the accuracy remains exponentially close to 1 in this regime.

## B. Discussion of Assumptions

Our derivation of Formula (1) required four assumptions that we discuss in turn.

1. In principle, our core assumption [**A1**] is falsifiable. As explained above, the output of the effective model can be empirically determined by running the LLM on all possible input sequences. It may then be checked whether this output can be generated by slightly varying the parameters of a potential idealized model that solves the task with perfect accuracy. Our empirical results suggest that assumption [**A1**] holds in several cases but also fails in at least one case — that of vanilla addition with Pro.

   Since state-of-the-art LLMs are based on the transformer architecture, it is natural to assume that in a specific setting, their operation can be modeled via a small effective transformer. However, one might ask whether the operation of the LLM could be modeled via some other effective sequence-to-sequence network. We leave this question to future work.

2. Our second assumption [**A2**] is clearly too simple to hold exactly in a more elaborate model. Even in an effective model, a more detailed output layer can be set up as follows. If the vector produced by previous layers in the model is $w$ then we posit that the model outputs a token $t_i$ with probability

$$p_i = \frac{e^{\beta w \cdot \mathcal{E}_{\text{emb}}(t_i)}}{\mathcal{Z}}; \qquad \mathcal{Z} = \sum_j e^{\beta w \cdot \mathcal{E}_{\text{emb}}(t_j)}; \tag{31}$$

   where $\mathcal{E}_{\text{emb}}(t_i)$ is the embedding vector corresponding to the token $t_i$, the sum over $j$ runs over all candidate tokens and $\beta$ is an inverse temperature.

   If $t_i$ is the correct expected token and $w = \mathcal{E}_{\text{emb}}(t_i) + \epsilon$, then the probability that the model will output the correct token depends with the inner product of $\epsilon$ with the embedding vectors. This probability is not an isotropic function of $\epsilon$ as assumed in [**A2**].

   These non-isotropic effects can be incorporated into our model at the cost of introducing additional parameters that contain information about the geometry of embedding vectors. We have chosen not to do so in order to obtain the simplest possible accuracy formula.

   Nevertheless, we note that assumption [**A2**] is responsible for an important feature of our accuracy formula: it suggests that the model is able to successfully correct errors until the complexity crosses a threshold beyond which the accuracy decays rapidly. Our empirical data clearly shows the presence of such an effect.

3. Our third assumption [**A3**] is the easiest to change. The incomplete gamma function in (1) arises from the cumulative distribution function of a Gaussian. Substituting the Gaussian in [**A3**] with a different distribution would lead to an accuracy formula involving the cumulative distribution function of that distribution.

   Our choice of the Gaussian is motivated by simplicity rather than deeper theoretical considerations. A more-detailed study of the length of the longest-error vector might suggest a different distribution, leading to an improvement of the formula (1).

4. If one accepts that errors can be understood via an effective transformer, as posited by [**A1**], then the scaling shown in ([**A4**]) is rather robust. A more interesting question has to do with the value of $\alpha$.

   As explained in Appendix A.3, apart from the choice $\alpha = 1$ that we have used, the choice $\alpha = \frac{1}{2}$ is also natural. Perhaps, further investigation will reveal that the larger values of $q$ that arise with that choice also have a simple interpretation. A third alternative is to allow $\alpha$ to vary, and to fit it empirically. We have not done so, to avoid introducing additional parameters in our model.

## C. Details about Graphs and Error Bars

The graphs in this paper show the accuracy of LLMs on various tasks as a function of a complexity parameter, $c$. For each value of $c$, we conduct a number of trials, denoted by $N$. On each trial, the model either gets the right answer or the wrong answer. Since the answer is well defined for every problem we consider, we do not use any notion of a "partially correct answer". If the number of right answers is $R$, the mean accuracy is set to

$$a = \frac{R}{N}. \tag{32}$$

We now plot error bars as follows. Let the true accuracy of the model be $p$. We have no prior information about this, and so our prior estimate is a flat probability distribution for $p$

$$P_{\text{prior}}(p) = 1, \qquad p \in [0, 1] \tag{33}$$

Given that $R$ correct answers were obtained in $N$ trials we now use Bayes theorem to obtain a posterior distribution

$$P_{\text{post}}(p|R, N) = \frac{P(R, N|p)P_{\text{prior}}(p)}{P(R, N)} = \frac{P(R, N|p)}{P(R, N)}. \tag{34}$$

Using the binomial distribution, we see straightforwardly that

$$P(R, N|p) = \frac{\Gamma(N + 1)}{\Gamma(R + 1)\Gamma(N - R + 1)} p^R (1 - p)^{N-R}. \tag{35}$$

Now, $P(R, N)$ can be fixed by demanding that the posterior probability distribution be normalized,

$$\int P_{\text{post}}(p|R, N)dp = 1. \tag{36}$$

Using this condition, we find $P(R, N) = \frac{1}{N+1}$, independent of $R$. Substituting this value in (34), together with (35), leads to

$$P_{\text{post}}(p|R, N) = \frac{\Gamma(N + 2)}{\Gamma(R + 1)\Gamma(N - R + 1)} p^R (1 - p)^{N-R}. \tag{37}$$

The cumulative distribution function for this posterior probability distribution is given by

$$C(p, R, N) = \int P_{\text{post}}(p|R, N)dp = \frac{\Gamma(N + 2)B_p(R + 1, N - R + 1)}{\Gamma(R + 1)\Gamma(N - R + 1)}, \tag{38}$$

where $B_p$ is the incomplete beta function.

In each graph, we mark 95% confidence intervals by solving for $\mu$ with the property that

$$C(\min(\frac{R}{N} + \mu, 1), R, N) - C(\max(\frac{R}{N} - \mu, 0), R, N) = .95. \tag{39}$$

This ensures that the observed value of the accuracy $\frac{R}{N}$ is always part of the confidence interval.

**Goodness of fit.**   It is also possible to define a measure of goodness of fit. Say that we have empirical data for $n$ distinct values of $c$. For each value of $c$, we denote the empirically measured accuracy by $a_c$, the accuracy predicted by Formula (1) by $\hat{a}_c$ and the size of the confidence interval computed using the prescription above by $\sigma_c$. We then define

$$\chi^2 = \frac{1}{n} \sum_c \frac{(a_c - \hat{a}_c)^2}{\sigma_c^2} \tag{40}$$

This is *not* an absolute measure of the goodness of fit, since $\sigma_c$ can always be reduced by taking more samples. Since our interest is not in the absolute value of $\chi^2$, it also does not matter that $\sigma_c$ is a 95% confidence interval rather than a single standard deviation. However, this measure can be used to compare two models as was done in Appendix A.4.

## C.1. Best Fit Parameters

| Task | Gemini Flash | | DeepSeek | | Gemini Pro | |
|---|---|---|---|---|---|---|
| | $r$ | $q$ | $r$ | $q$ | $r$ | $q$ |
| R | $(2.67 \pm 0.04) \times 10^{-4}$ | $4.2 \pm 0.2$ | $(2.27 \pm 0.04) \times 10^{-4}$ | $4.2 \pm 0.2$ | $(5.0 \pm 0.1) \times 10^{-5}$ | $1.8 \pm 0.1$ |
| NLT | $(9.64 \pm 0.07) \times 10^{-5}$ | $13.9 \pm 0.6$ | $(5.0 \pm 0.1) \times 10^{-4}$ | $2.8 \pm 0.1$ | $(5.7 \pm 0.2) \times 10^{-5}$ | $1.01 \pm 0.07$ |
| DP | $(1.30 \pm 0.01) \times 10^{-4}$ | $11.1 \pm 0.5$ | $(3.85 \pm 0.07) \times 10^{-4}$ | $4.2 \pm 0.2$ | $(1.47 \pm 0.01) \times 10^{-4}$ | $13.6 \pm 0.7$ |
| TH | $(2.61 \pm 0.05) \times 10^{-4}$ | $3.2 \pm 0.1$ | $(8.8 \pm 0.2) \times 10^{-4}$ | $3.4 \pm 0.1$ | $(7.5 \pm 0.1) \times 10^{-5}$ | $3.0 \pm 0.1$ |
| VA | $(4.14 \pm 0.07) \times 10^{-3}$ | $1.54 \pm 0.03$ | $(3.89 \pm 0.08) \times 10^{-3}$ | $2.33 \pm 0.07$ | $(1.1 \pm 0.1) \times 10^{-3}$ | $0.24 \pm 0.01$ |
| AA | $(1.69 \pm 0.03) \times 10^{-3}$ | $3.5 \pm 0.1$ | $(9.8 \pm 0.2) \times 10^{-4}$ | $3.1 \pm 0.1$ | $(2.2 \pm 0.2) \times 10^{-3}$ | $0.56 \pm 0.03$ |
| BA | $(7.84 \pm 0.08) \times 10^{-3}$ | $9.5 \pm 0.5$ | $(4.50 \pm 0.07) \times 10^{-3}$ | $5.2 \pm 0.3$ | $(5.16 \pm 0.06) \times 10^{-3}$ | $8.1 \pm 0.4$ |
| M | $0.0220 \pm 0.0007$ | $2.5 \pm 0.1$ | $0.0168 \pm 0.0003$ | $5.5 \pm 0.3$ | $(7.9 \pm 0.1) \times 10^{-3}$ | $4.8 \pm 0.3$ |

*Table 2.* Best fit parameters for various tasks. The tasks are indicated using the follows keys: R–Reversal; NLT-Nested Linear Transformations; DP-Dynamic Programming; TH-Tower of Hanoi; VA-Vanilla Addition; AA-Algorithmic Addition; BA-Binary Addition; M-Multiplication.

The improved prompt of section 4 was tested only with Flash. The best fit parameters for polynomial multiplication in Figure 9 are $r = (3.82 \pm 0.06) \times 10^{-3}$ and $q = 4.6 \pm 0.3$.

The errors in the parameters above indicate one standard deviation.

## C.2. Comments about Plots

In some graphs, points with $c$-values differing by 1 were collapsed into a single $c$-value for uniformity, and to economize computational costs. We list these cases below. Since our dataset is openly available, the reader may undo these transformations using the instructions below and plot the raw data rather than the adjusted data.

1. **Nested linear transformations:** In the graph depicting the accuracy of Flash, we always take $c = 5j + 3$ for integer $j$. This required adjusting $c \to c - 1$ in some cases where part of the data was inadvertently taken for $c$-values corresponding to $5j + 4$. This adjustment was not required for the graphs displaying the accuracy of DeepSeek or Pro. To undo this transformation in the dataset, one simply needs to set $c$ to be the length of the list.

2. **Vanilla addition:**

   (a) In the graph depicting the accuracy of Flash, $c$ is taken to be even for $c > 20$. This required adjusting $c \to c + 1$ in some cases since data was taken for odd values of $c$ as well. This adjustment was not required for the graphs displaying the accuracy of DeepSeek or Pro. To undo the transformation, one simply needs to set $c$ to be the length of the numbers to be added.

   (b) The example given to the model as part of the prompt, used for initial experiments with Flash and Pro, contained a typo. We emphasize that such errors do not have any bearing on our analysis, which does not assume that the prompt should be "clear". The prompt with a typo also defines an "effective model". However, we corrected the typo after noticing it and we have combined the datasets in the displayed graph. Empirically we found that the accuracy of both models was similar with or without the incorrect example, suggesting that the models were able to ignore this error. Perhaps this is because addition is a task encountered during training, where an in-context example does not affect the model's accuracy significantly. However, the dataset for these models includes a flag "Typo", which can be used to distinguish between data taken with the typo (flag = 1) and without the typo (flag = 0). The dataset for DeepSeek corresponds entirely to the prompt without the typo.

3. **Algorithmic addition:** $c$ is always taken to be even. This requires adjusting $c \to c + 1$ in some cases. These adjustments were made for all three graphs.

**Unclear output.** As already discussed, modern LLMs follow instructions accurately for the simple tasks tested here. Our prompts instructed the models to output their answers in a specific format. In about 99% of our examples — more specifically, in 200335 out of 203019 test prompts — we were able to successfully parse the model's output using automated methods. The remaining 1% of cases were discarded while preparing the final graphs.

# D. Prompts

### D.1. List Reversal

```
Given a list, you must write out its elements in reverse order individually. The
↪  list will be given in the format
LIST=[...];
And the output should be
R[0]=last entry;
R[1]=second-last entry;
...
R[len-1]=first entry;
where len is the length of the list. Note each reversed list element is on its
↪  own line. For example if the input had been
LIST=[2,3,5,7];
the expected output would be
R[0]=7;
R[1]=5;
R[2]=3;
R[3]=2;
Do not neglect to use the precise output format as shown in the example. You are
↪  not expected to use code or produce code snippets in the output. The list is
LIST=[9, 0, 4, 8, 1, 2, 8]
```

### D.2. Nested Linear Transformations

```
We have two lists of numbers, LIST1 and LIST2, and an initial value CHAIN[0].
↪  Your task is to propagate the initial value through the lists as follows. At
↪  each step generate CHAIN[i+1]=CHAIN[i]*LIST1[i]+LIST2[i] i.e. multiply the
↪  current value with an element of the first list and add the element of the
↪  second list to generate the next value. The process stops when the lists are
↪  exhausted. Write the chain of digits obtained as
CHAIN=[CHAIN[0],CHAIN[1],...];
For instance, if the initial value had been CHAIN[0]=3 and the lists had been
↪  LIST1=[1,2] and LIST2=[3,4] you would calculate CHAIN[1]=1*3 + 3 = 6;
↪  CHAIN[2]=2*6+4=16 and the final output would be
CHAIN=[3,6,16];
Perform this procedure with the initial value
CHAIN[0]=2
and the lists
LIST1=[9, 0, 1, 3]
LIST2=[-9, 1, 5, -9]
Do not neglect to use the keyword CHAIN, and the precise output format as shown
↪  in the example. You are not expected to use code or produce code snippets in
↪  the output.
```

### D.3. Dynamic Programming

```
Given the following list of non-negative integers
LST=[1, 5, 8, 7, 5];
you must find the subsequence with the largest sum that has no adjacent elements.
↪  If two subsequences have the same sum, choose the one that is
↪  lexicographically smaller in terms of list indices (not list elements).
↪  Output a list of the same length as LST in the format
ANSWER=[...];
which contains 1 for each chosen index and 2 for each index that is not chosen.
```

```
For example, if the input had been
LST=[8,0,6,9];
the expected output would be
ANSWER=[1,2,2,1];
This is a dynamic programming problem that can be solved recursively by means of
↪   the following function given in pseudocode.

function choose_non_adjacent(LST):
  n = length of LST
  // dp[i] will store the maximum sum obtainable from the subarray LST[i:]
  // choose[i] will be true if we select LST[i] in the optimal solution for
  ↪   LST[i:]
  dp = array of size (n + 2) initialized to 0
  choose = array of size n initialized to false

  // Iterate from the end of the list to the beginning
  for i from n - 1 down to 0:
    // Option 1: Take the current element. We cannot take the next one (i+1).
    // The best sum would be LST[i] + the best sum from LST[i+2:]
    take = LST[i] + dp[i + 2]

    // Option 2: Skip the current element.
    // The best sum would be the best sum from LST[i+1:]
    skip = dp[i + 1]

    // Compare the two options. In case of a tie, we prefer to take the current
    ↪   element.
    if take >= skip:
      dp[i] = take
      choose[i] = true
    else:
      dp[i] = skip
      choose[i] = false

  // Reconstruct the result array based on the 'choose' array
  res = array of size n initialized to 2
  i = 0
  while i < n:
    // If the optimal choice was to take element i
    if choose[i]:
      res[i] = 1
      // If we take i, we must skip i+1, so we can jump ahead by 2
      i = i + 2
    else:
      // If we skip i, we move to the next element
      i = i + 1

  return res
```

Do not neglect to use the keyword ANSWER and the precise output format as shown
↪   in the example. You are not expected to use code or generate code snippets in
↪   the answer.

## D.4. Tower of Hanoi

```
Consider the Tower of Hanoi problem with 10 disks starting at tower 0 that need
↪  to be moved to tower 1. The disks are labelled (smallest to largest) by the
↪  list of numbers
DISK_LABELS=[0, 7, 8, 4, 3, 6, 1, 2, 9, 5].
This means that 0 is the smallest disk, 7 is the second-smallest etc. The
↪  starting position of all the disks is 0, which can be represented by an
↪  array: pos[0]=0; pos[1]=0; ... To get the n^th move, we represent n as a
↪  binary number. The position of the least significant 1 that appears in the
↪  binary representation tells us the index of the  disk that needs to be moved.
↪  Once the index of the disk is known, the corresponding position must be
↪  updated as follows. The smallest disk always cycles between the towers as
↪  0->2->1->0... and the second smallest as 0->1->2->0... and so on. The disk
↪  labeled by DISK_LABELS[i] for even i cycles like the smallest disk, and for
↪  odd i it cycles like the second smallest disk. Your task is to output the
↪  first 30 moves in the form
ANSWER=[move1, move2...];
where each move is itself a tuple with exactly 3 entries
↪  (disk-to-be-moved,starting-tower,ending-tower).
For example, if there had been 4 disks labelled by
DISK_LABELS=[0, 3, 1, 2]
and you had been asked to output the first 2 moves, the expected output would be
ANSWER=[(0, 0, 2), (3, 0, 1)];
Do not neglect to use the keyword ANSWER and the precise output format as shown
↪  in the example. You are not expected to use code or generate code snippets in
↪  the answer
```

## D.5. Vanilla Addition

```
Calculate 684041602 + 386049129. Output the final answer in the form ANSWER:
↪  <answer>. For example if the numbers had been 34 and 59 the expected output
↪  would have been ANSWER: 93. Do not neglect to use the keywork ANSWER:
```

## D.6. Addition with Algorithm

```
I want to manipulate two numbers: 7212208817, 1549886112 to get a third number.
↪  Use the following algorithm. Output the answer to each step of the algorithm
↪  using the exact keywords given below.
1) Break each number into a list of digits from left to right. Output the answer
↪  as "ANSLIST1:<list1>" and "ANSLIST2:<list2>"
2) Reverse each list by enumerating its elements from right to left.  Output the
↪  answer as "ANSREVLIST1:<reversed list1>" and "ANSREVLIST2:<reversed list2>"
3) Zip the elements of the two reversed lists by pairing entries to make a list
↪  of tuples. If the digits in the shorter list run out, use 0 for the remaining
↪  entries. Output the answer as "ANSPAIRLIST:<paired list>"
4) Add the elements of each tuple together to obtain a list of integers. Output
↪  the answer as "ANSSUMSLIST:<sums list>"
5) Convert this list of integers into a list of single digits as follows.
↪  Proceeding from left to right, keep the units place for each entry, and move
↪  the tens place into a carry that is added to the next entry. Repeat this for
↪  all entries. If the last entry has a nonzero carry, append it as an
↪  additional element to the list. Output the answer as
↪  "ANSDIGITSLIST:<digitslist>"
6) Reverse this list of digits by enumerating its elements from right to left.
↪  Output the answer as "ANSREVDIGITSLIST:<reversed digits list>"
```

7) Assemble this reversed list into a number by concatenating the entries from
↪ left to right. Output the answer as "ANSNUM:<number>".
For example, if the numbers are 123 and 4567, the expected output is
ANSLIST1: [1,2,3]
ANSLIST2: [4,5,6,7]
ANSREVLIST1: [3,2,1]
ANSREVLIST2: [7,6,5,4]
ANSPAIRLIST: [(3,7), (2,6), (1,5), (0,4)]
ANSSUMSLIST: [10, 8, 6,4]
ANSDIGITSLIST: [0,9,6,4]
ANSREVDIGITSLIST: [4,6,9,0]
ANSNUM: 4690
Do not neglect to use the keywords and output format precisely as given in the
↪ example above. You are not expected to use code or generate code snippets in
↪ the answer.

### D.7. Binary Addition

Add the two binary numbers 1100001010001111100001 + 101000001101000101011.
↪ Output the final answer as ANSWER: <answer>. For example if the numbers are
↪ 1010 and 100 the expected output is ANSWER: 1110. Do not neglect to use the
↪ keywork ANSWER:

### D.8. Multiplication

Calculate 7869*85201343475254159272 using the following algorithm.
(1)Compute subproducts by multiplying each digit of the smaller number with the
↪ larger number and adding zeros as appropriate to the place value. Output the
↪ list of subproducts as
SUBPRODLIST=[<list of subproducts>];
(2)Add the subproducts to obtain the final answer.
Output the final answer as a single element list
ANSWER=[<answer>];
For example, if the problem had been 12*365 the expected output would have been
SUBPRODLIST=[730, 3650];
ANSWER=[4380];
Do not neglect to use the given keywords, and output format exactly as shown in
↪ the example. You are not expected to use code or generate code snippets in
↪ the answer.

### D.9. Multiplication using Intermediate Polynomials

Given two numbers, you must produce another number using intermediate polynomials
↪ as detailed in the following algorithm.
(1) Convert the two numbers into polynomials, P(x) and Q(x), by setting the nth
↪ digit (i.e. the coefficient of 10^n) to be the coefficient of x^n. Output the
↪ coefficients of the polynomials as
P0=coefficient of x^0 in P(x);
P1=coefficient of x^1 in P(x);
...
Pn=coefficient of x^n in P(x);
Q0=coefficient of x^0 in Q(x);
Q1=coefficient of x^1 in Q(x);
...
Qn=coefficient of x^n in Q(x);

This first step is error prone and so check the conversion *carefully* to make
↪ sure that digits from either number are not omitted or repeated.
(2) Multiply the polynomials to get R(x)=P(x)*Q(x). The coefficients, Rk are
↪ obtained by summing all Pn*Qm with n+m=k
Output the coefficients of R(x) as:
R0=coefficient of x^0 in R(x);
R1=coefficient of x^1 in R(x);
...
Rn=coefficient of x^n in R(x);
(3) Now transform R(x) to a new polynomial S(x) as follows. Starting with carry=0
↪ define S0=(R0+carry) mod 10 and the new carry = floor((R0+carry)/10).
↪ Continue this way for S1, S2 .. and so on up till Sq where q is the degree of
↪ R(x). If the last carry is not zero, add additional terms S{{q+1}}, S{{q+2}},
↪ until the carry is zero. Output the elements of S(x) as
S0=coefficient of x^0 in S(x);
S1=coefficient of x^1 in S(x);
...
Sn=coefficient of x^n in S(x);
(4) Convert S(x) back to a number whose nth digit is Sn. Ourput the number as
ANS=number produced;
This last step is also prone to errors, so make sure to check the conversion
↪ *carefully* and ensure that digits are not omitted or repeated.
For example, if the input is 34 and 25, the expected output is
P0=4;
P1=3;
Q0=5;
Q1=2;
R0=20;
R1=23;
R2=6;
S0=0;
S1=5;
S2=8;
ANS=850;
Regardless of whatever intermediate calculations you perform, make sure to output
↪ the numerical answer for each polynomial coefficient and the final answer
↪ *separately* as shown above --- 'P0=4;' --- with a newline, keyword for the
↪ coefficient or answer, an equal sign, a numerical value and a semicolon
↪ followed by a newline. Do not attempt to to shortcut the process and follow
↪ the algorithm precisely. Do not produce code or code snippets in the output.
The numbers are
7869 and
611912436665956692;

