# OpenReview forum: "A model of errors in transformers"
_ICML.cc/2026/Conference — ICML 2026 regular_

### Official Review · Reviewer_nh8f · 2026-03-04

**Soundness:** 3
**Presentation:** 3
**Significance:** 3
**Originality:** 3
**Overall Recommendation:** 4
**Confidence:** 4

**Summary:**

The authors proposes a error modeling framework that casts complex Transformer models as an effective model with small parameter perturbations. It assumes that errors arise from slight deviations in the attention mechanism, which accumulate over long sequences and eventually lead to incorrect predictions once a certain threshold is exceeded. Based on this perspective, the authors introduce two key parameters (the noise rate $r$ and the number of error directions $q$), and derive an analytical expression characterizing the relationship between model accuracy and task complexity.

The validity of the proposed formulation is supported by extensive experiments. The results show that the derived formula aligns well with empirical error trends in most cases. The authors also examine instances where the model deviates from theoretical predictions and suggest that such discrepancies may be due to the adoption of different strategies at varying levels of task complexity. Overall, the paper offers a low-parameter, physically motivated error model that provides a useful perspective for understanding and predicting the performance of large language models on long-sequence reasoning tasks.

**Compliance With Llm Reviewing Policy:**

Affirmed.

**Final Justification:**

The original submission lacked a systematic sensitivity analysis of the key parameter $\alpha$. The authors provided preliminary results in their rebuttal and committed to including a more thorough analysis in the revised version, which partially alleviates concerns; however, the evaluation in the original submission remains insufficient. I will maintain my score.

**Key Questions For Authors:**

1. I suggest explicitly discussing the limitations of the method.

2. The appendix only presents results for $\alpha=1$ and $\alpha=\frac{1}{2}$, but does not include a systematic sensitivity analysis of $\alpha$.
How would other values affect the theoretical predictions and the quality of experimental fitting?

**Limitations:**

yes

**Strengths And Weaknesses:**

Strengths:

1. The focus on the error mechanisms of LLM in long-sequence reasoning carries substantial research merit.

2. The experimental design covers multiple structured reasoning tasks and is validated across several representative models, providing solid empirical support for the proposed theory.

3. The paper derives an analytical expression that characterizes the relationship between model accuracy and task complexity under the proposed error accumulation framework.

4. The paper is well written and relatively easy to follow.

Weaknesses:

1. Although the limitations of the method are discussed, the treatment is not sufficiently focused or systematic.

2. The rationality of sharing the same error model across different tasks also requires further theoretical justification.

3. This paper lacks a finer-grained parameter sensitivity analysis.

---

> ### Author Rebuttal · Authors · 2026-03-30
>
> We would like to thank the reviewer for helpful comments.
>
> **Response to reviewer's note on weaknesses.**
>
> > Although the limitations of the method are discussed, the treatment is not sufficiently focused or systematic.
>
> Discussion of limitations: We would be glad to add additional discussion of the limitations of our method. As we mentioned in our response to reviewer nh8f, our methodology is inspired by physics where it is traditional to examine a complex system in the simplest possible setting in order to unearth broad underlying principles. However, this is also a limitation of our study since our model applies only to those tasks where the model is required to process and output a low-entropy deterministic sequence. Our two-parameter model provides a good description for a remarkable range of such tasks but the empirical fit can be improved. In addition, as we mentioned to other reviewers, additional experiments – specifically studying the error rate of models that (1) utilize some other form of sequence-to-sequence mapping instead of the attention mechanism e.g., LSTMs, and (2) attention-only models which do not have any feed forward layers –  can help to clarify some aspects of our model.
>
> > The rationality of sharing the same error model across different tasks also requires further theoretical justification.
>
> Same error model across tasks: We would like to reiterate our perspective. In the natural sciences, and especially physics, it is traditional to examine simplified settings. But we still look for a theory that can unify a broad range of phenomena.
>
> It is noteworthy that our theory describes the error rate of three state-of-the-art LLMs over so many tasks. We see this as a success, and it suggests that we have been able to accurately identify at least some underlying common causes for errors made by LLMs.
>
> > This paper lacks a finer-grained parameter sensitivity analysis.
>
> Finer-grained sensitivity analysis: We direct the reviewer to Appendix C. Here, we devoted a considerable amount of attention to error analysis and the uncertainty in our parameters; in our opinion this analysis goes considerably beyond the level of error analysis that is conventionally presented in the AI/ML literature.
>
> Moreover, our raw data, comprising the results of about 200K prompts, was included with the submission and will be available openly for other researchers if they wish to perform additional statistical analyses.
>
>
> **Response to questions**
>
> - Limitations: Please see point 1 in the first section of our reply to this review.  Following the reviewer's suggestion, we would be glad to include additional discussion in a revised version of our paper.
>
> - Value of $\alpha$: We fixed the parameter $\alpha$ in order to keep the number of variable parameters as small as possible and we only discussed two possibilities: $\alpha = 1/2$ which is natural when the errors in the attention mechanism are uncorrelated and $\alpha = 1$, which is natural when the are correlated. The goodness-of-fit for these choices is comparable although the values of $q,r$ vary with $\alpha$. Following the reviewer's suggestion, we would be glad to include additional analysis of the parameter $\alpha$ in a revised version and show how the goodness-of-fit and the values of $q,r$ vary with $\alpha$.

---

> > ### Author Rebuttal · Reviewer_nh8f · 2026-04-03
> >
> > The lack of $\alpha$ sensitivity analysis constitutes a notable weakness that undermines confidence in the method’s robustness.
> >
> > Thank you to the authors for your response!
> > I will keep my score.

---

> > > ### Author Response · Authors · 2026-04-06
> > >
> > > In Table 1 in Appendix A.4, we present the goodness of fit and the values of q and r for alpha = ½ and 1 for the nested linear transformation task. In addition to this, in our previous message, we promised to include an $\alpha$-sensitivity analysis in a revised version. While we are unable to upload a revised version at this stage, we have performed this analysis and can indicate the key results in words here.
> > >
> > > The goodness-of-fit can be determined using  the $\chi^2$-variable as defined in Appendix C. We define $\chi^2_{\rm{mean}}$ by averaging over all 24 experiments in section 3 (8 experiments and 3 models). We find that $\alpha$ values below 0.25 yield a very large value of $\chi^2_{\rm{mean}}$ indicating a poor fit. At $\alpha = 0.25$, we have $\chi^2_{\rm{mean}}=0.71$. This value declines steadily and at $\alpha=0.5$, we find $\chi^2_{\rm{mean}}=0.49$. Please recall that the value of $\alpha=0.5$ corresponds to uncorrelated errors in the effective model in our analysis and was one of the extreme (but plausible) values that we discussed.
> > > The value of $\chi^2_{\rm{mean}}$ then stays relatively flat, as expected but it attains a minimum at $\alpha=0.715$ with $\chi^2_{\rm{mean}} = 0.48$. It then rises slightly and at $\alpha = 1$, which was the value used in the main text, we find that $\chi^2_{\rm{mean}}=0.49$, which is the same as its value at $\alpha=0.5$. The value of $\chi^2_{\rm{mean}}$ continues to rise indicating that the fit becomes poor and at $\alpha = 2$, we find $\chi^2_{\rm{mean}}=0.61$.
> > >
> > > This is in line with the expectations expressed in the paper. In particular, the “best fit” for the experiments considered here is obtained by studying errors that are partly correlated and partly uncorrelated as the scaling of $\alpha=0.715$ would indicate. This value was determined by numerical minimization, and depends on the specific experiments that we conducted. It might be different for other experiments. The value of $\alpha = 1$, which yields itself to a much simpler interpretation yields a comparably good fit, and so we would suggest using this value to avoid overfitting.
> > >
> > > As expected, and as we indicated in our previous reply, the values of $q,r$ depend more strongly on $\alpha$ than the goodness of fit. This can also be seen in Table 1 for alpha = ½ and 1. They both decline monotonically with $\alpha$ in our experiments. As we mentioned, the values of $q$ for $\alpha = 1$ are most easily interpretable.
> > >
> > > In a revised version, we will include graphs showing the variation of the values of $q$ and $r$ for all experiments with $\alpha$ and also the variation of $\chi^2_{\rm{mean}}$ with $\alpha$.
> > > We would like to thank the reviewer for encouraging us to perform this additional analysis. We hope that this resolves any lingering concerns that the reviewer might have about the robustness of our analysis.

---

### Official Review · Reviewer_oQTM · 2026-03-10

**Soundness:** 2
**Presentation:** 3
**Significance:** 2
**Originality:** 3
**Overall Recommendation:** 4
**Confidence:** 2

**Summary:**

This paper investigates the degradation of Large Language Model (LLM) performance on deterministic, repetitive tasks (such as arithmetic, list reversal, and dynamic programming) as the complexity or context length increases. The authors propose a mathematical model to predict this error rate, drawing inspiration from "effective field theory" in physics. By making several assumptions about how small errors in the attention mechanism accumulate, they derive a two-parameter formula. The authors then conduct extensive empirical evaluations using Gemini 2.5 Flash, Gemini 2.5 Pro, and DeepSeek R1 across eight tasks, demonstrating that their formula closely matches the empirical accuracy decay curves in most cases, and they offer a prompt-engineering strategy to mitigate these errors.

**Compliance With Llm Reviewing Policy:**

Affirmed.

**Final Justification:**

While I appreciate the authors' physics-inspired "spherical cow" defense, my core concerns remain partially unresolved. The heuristic nature of the mathematical assumptions (which still feel largely like post-hoc curve fitting) and the narrow focus on toy algorithmic tasks are fundamental methodological choices. Consequently, I maintain some reservations about the soundness of the first-principle derivations and the broader significance of this framework for real-world, semantic LLM applications.

However, the originality of this phenomenological approach and the clarity of the presentation are excellent. The sheer scale of the empirical validation, testing over 200,000 prompts across modern models, reveals a striking and highly consistent pattern of failure.

Weighing these dimensions, I believe the exceptionally strong empirical groundwork and the novel perspective outweigh the theoretical limitations. Even if treated primarily as an empirical baseline, the community might benefit from this work. Therefore, the rebuttal has positively influenced my overall evaluation, and I have decided to raise my score from a 3 to a 4.

**Key Questions For Authors:**

1. Can you provide a more rigorous, mechanistic justification for the choice of a Gaussian distribution [A3] and the specific scaling factor $\alpha=1$ [A4], independent of the fact that they yield good empirical fits?
2. Is there a principled way within your proposed framework to predict when an LLM will violate assumption [A1] (e.g., changing its underlying algorithm, as seen with Gemini Pro), or can this only be diagnosed post-hoc after the empirical data fails to fit the curve?
3. How might this "effective field theory" approach be adapted or tested for tasks that are not strictly deterministic or token-repetitive (e.g., standard question answering, summarization, or logic puzzles over long contexts)?

**Limitations:**

yes

**Strengths And Weaknesses:**

## Strengths:
1. The authors strive to address a relevant issue by seeking a quantitative understanding of why and how LLMs fail on long, repetitive contexts, which is a widely recognized limitation of current models.
2. The paper considers a general aspect of transformer behavior (error accumulation over discrete reasoning steps) through a creative, interdisciplinary lens inspired by physics.
3. The empirical evaluation is thorough. Testing 200,000 distinct prompts across multiple state-of-the-art models and various algorithmic tasks provides a solid dataset for the community.

## Weaknesses:
1. The theoretical derivation appears somewhat heuristic and risks being a post-hoc curve-fitting exercise. Specifically, assumptions [A2], [A3], and [A4] (such as modeling the error coefficients with a Gaussian distribution and fixing the scaling parameter $\alpha=1$) feel tailored to match the empirical data rather than derived from first principles.
2. The scope of the tasks is narrow. The evaluation is restricted entirely to toy deterministic algorithmic tasks (like reversing lists or Tower of Hanoi). It is unclear if this mathematical model has any bearing on the typical use cases of LLMs, such as natural language understanding, open-ended reasoning, or coding, limiting the work's broader significance.
3. The framework seems to lack predictive power for edge cases. For instance, when the formula completely fails for Gemini 2.5 Pro on vanilla addition, the authors hypothesize that the model might be using inconsistent algorithms at different lengths. However, the proposed framework offers no a priori way to predict when such assumption violations will occur, reducing its practical utility.

---

> ### Author Rebuttal · Authors · 2026-03-30
>
> We would like to thank the reviewer for helpful comments. Before we respond to specific points, we would like to make a general comment on perspective.
>
> This paper adopts the perspective used in physics and other natural sciences. When faced with a complex system, we first choose a simple and clean setting in which the system's behavior can be quantified as easily as possible. We then model the system using semi-empirical assumptions and then check if the predictions of the model match the system's response. This method has been enormously successful in understanding nature and uncovering general principles that underlie natural systems. We believe that it should also be applied to the study of modern LLMs, and this paper is an attempt to initiate such a program.
>
> **Response to reviewer's note on weaknesses**
> > The theoretical derivation appears somewhat heuristic and risks being a post-hoc curve-fitting exercise. Specifically, assumptions [A2], [A3], and [A4] ... feel tailored to match the empirical data rather than derived from first principles.
>
> We provide a detailed explanation of assumptions A2 and A4, trying to motivate them from first principles. Assumption A2 is justified on the basis of the output layer which projects from the embedding space back to the space of tokens.  An incorrect token is expected to be predicted only if the length of the error is comparable to the distance (in the embedding space) between the correct token and its neighbors. The precise threshold depends on the detailed geometry of the embeddings. However since we are concerned with the average accuracy over all possible input sequences, from a mean field perspective, it is reasonable to assume that an error is made when the length of the error vector exceeds a threshold. This is all the assumption A2 states.
>
> We justify Assumption A4 in Appendix A.3. The main observation is that errors accumulate across the attention mechanism. If these errors are correlated (which is reasonable when the number of tokens is much smaller than the sequence length) we expect \alpha = 1. If these errors are uncorrelated, we expect \alpha = 1/2.
>
> We already mention in the paper that Assumption A3 is motivated by simplicity rather than deeper theoretical considerations. The Gaussian is a simple and generic distribution and also fits the empirical data well. The difficulty in justifying this from a first-principles perspective is that we do not currently know how to relate the parameters of the effective model from the parameters of the LLM. Hopefully, in the future, as the theory advances this will be more practicable. We would be glad to emphasize this further in a revised version.
>
> Finally, we would like to remind the reviewer of our perspective noted above; semi-empirical assumptions are crucial in obtaining a simplified description of a complex system.
>
> > The scope of the tasks is narrow. The evaluation is restricted entirely to toy deterministic algorithmic tasks (like reversing lists or Tower of Hanoi)...
>
> We deliberately made a choice to focus on a narrow set of tasks. This is related to our perspective described above. When faced with a complex system, it is very important to examine its behavior in a simplified and clean setting since these analyses help in uncovering deeper principles about the system. A common joke about physicists --- albeit one that contains a significant dose of truth --- is that they look for "spherical cows". One can think of the simple set of tasks that we examine as "spherical cows".
>
> Of course, eventually we would like to apply the understanding that we glean from such systems to more-realistic scenarios. Based on the experience of the natural sciences, we are very optimistic that the program we seek to initiate in this paper will eventually bear such fruits.
>
> > The framework seems to lack predictive power for edge cases. For instance, when the formula completely fails for Gemini 2.5 Pro on vanilla addition, the authors hypothesize that the model might be using inconsistent algorithms at different lengths. However, the proposed framework offers no a priori way to predict when such assumption violations will occur, reducing its practical utility.
>
> We again remind the reviewer of the perspective explained above. In physics, when one finds a model that applies to a range of phenomena, edge cases are a matter of great interest since they point to undiscovered effects. It is not possible to a priori predict such phenomena.
>
> The primary objective of our paper is to improve our understanding of LLMs and not immediate "practical utility" although we believe that improved understanding does eventually lead to practical applications.
>
> **Response to questions**
> - Assumption A3 and A4: Please see point (1) in the "response to weaknesses" above.
> - Detecting violations of A1: Please see point (3) in the section above.
> - Other tasks: Please see point (2) in above.

---

> > ### Author Rebuttal · Reviewer_oQTM · 2026-04-03
> >
> > I thank the authors for their detailed rebuttal and for clarifying their philosophical perspective.
> >
> > I selected option (c) because my core concerns regarding the heuristic, post-hoc nature of the mathematical assumptions (e.g., the Gaussian distribution and scaling choices) and the narrow focus on toy algorithmic tasks represent fundamental limitations of the paper's methodology. As the authors acknowledge, these are deliberate design choices. Therefore, these theoretical limitations cannot be simply "fixed" in a short rebuttal, leaving my concerns about the framework's predictive power for real-world tasks unresolved.
> >
> > Despite this, I have decided to raise my score from a 3 to a 4. The authors strive to address a relevant issue by modeling how LLM performance degrades on longer sequences. Furthermore, the paper considers a general aspect of error accumulation in transformers. Even if I do not fully agree with the "physics-inspired" defense and view the mathematical framework primarily as a phenomenological curve-fit rather than a first-principle derivation, the empirical contribution is undeniably strong. Testing over 200,000 prompts across multiple modern models reveals a highly consistent and striking pattern of failure. This large-scale empirical groundwork and the novel perspective are valuable enough that the community will benefit from discussing them at the conference.

---

> > > ### Author Response · Authors · 2026-04-06
> > >
> > > We would like to thank the reviewer for their encouraging comments, and for the revised  score. We agree that our theoretical analysis relies on empirically inspired intuition, and applies to a specific set of tasks. We also agree that it would be very desirable to have some simple set of “principles” from which we could develop a theory of LLMs that is applicable to a much broader set of tasks.
> > >
> > > We feel that such efforts to develop a “theory of LLMs” are important but currently somewhat neglected in the field. The current paper aims to contribute to this theoretical program, and we were glad that we could derive a simple two-parameter formula that relates our empirical results to a small set of  assumptions. At the moment, our efforts are incipient in nature but we hope that the framework and results in this paper will contribute to a broader theory that can explain the workings of LLMs from a set of fundamental principles.

---

### Official Review · Reviewer_Ah8A · 2026-03-13

**Soundness:** 3
**Presentation:** 3
**Significance:** 3
**Originality:** 4
**Overall Recommendation:** 4
**Confidence:** 3

**Summary:**

This article proposes a simple theoretical model to explain why transformer-based LMs sometimes fail on long or complex tasks. The authors argue that small internal errors in attention and representation accumulate during generation, eventually becoming large enough to cause the model to output an incorrect token. They derive a mathematical formula predicting how task accuracy decreases as task complexity increases, using only two parameters. The model is tested across several algorithmic tasks, such as arithmetic, list manipulation, and the Tower of Hanoi, and shows good agreement with empirical results from multiple LLMs. Overall, the work suggests that many reasoning failures in transformers arise from noise accumulation rather than a lack of reasoning capability.

**Compliance With Llm Reviewing Policy:**

Affirmed.

**Final Justification:**

The “physics-inspired” argument suggests that the theory is not yet applicable to realistic settings or larger vocabularies, and the failure cases observed in Gemini Pro indicate that the model’s assumptions may be overly rigid. Hence, I would prefer to keep my initial score of 4 (weak accept).

**Key Questions For Authors:**

1. What determines the parameter q? Can it be predicted based on the vocabulary structure or specific properties of the task?
2. What additional experiments could help verify that the observed failures are caused by attention errors accumulating over multiple steps? For example, do you observe a monotonic increase in representation variance as the sequence length grows?
3. If errors indeed accumulate through the attention mechanism, what types of architectural or training modifications might mitigate this issue?
4. Section 3.5 suggests that Gemini Pro may switch between different algorithms depending on the task length. Is there a way to validate this behavior and identify when such algorithm changes occur?

**Limitations:**

Refer to the weaknesses.

**Strengths And Weaknesses:**

**Strengths**
1. The authors address an important question: why the accuracy of LLMs decreases on long, deterministic tasks such as arithmetic or algorithmic reasoning. The work examines a general property of transformer behavior rather than focusing on a single benchmark.
2. The proposed error model reduces the complexity of LLMs, despite their billions of parameters, to just two key parameters, allowing a closed-form prediction of accuracy as a function of task complexity.
3. The model fits the empirical accuracy curves well across most of the evaluated tasks.
4. The paper suggests that failures arise from the accumulation of small attention errors rather than from a lack of reasoning ability or limitations in the architecture. This perspective challenges common assumptions and provides a plausible mechanistic explanation.
5. The results also demonstrate that the theory can guide prompt design to improve model performance.

**Weaknesses**
1. The model relies on several strong assumptions that may not always hold in practice; for example, the authors note a failure case in the vanilla addition task for Gemini Pro.
2. All the tasks examined in this study share a similar algorithmic structure and use a small vocabulary, which may limit the model’s ability to generalize to other domains such as commonsense reasoning or factual question answering.
3. The paper does not provide baseline models or alternative approaches for comparison with the proposed method.

---

> ### Author Rebuttal · Authors · 2026-03-30
>
> We would like to thank the reviewer for helpful comments.
>
> **Response to reviewer's note on weaknesses.**
>
> > The model relies on several strong assumptions that may not always hold in practice; for example, the authors note a failure case in the vanilla addition task for Gemini Pro.
>
> We believe that it is helpful to adopt a physics-inspired perspective to study the behavior of LLMs. The idea is to construct a simple model for how the LLM functions and the cause of errors, which abstracts away from irrelevant details.  Using this perspective we are able to reduce the enormous complexity of modern LLMs to just two parameters for a variety of tasks,  which, as the reviewer also notes, is a remarkable simplification.
>
> In physics, the cost that one pays for such simplification is that the resulting model  is not universally applicable. For example, the equations of fluid dynamics are obtained by making specific assumptions about a state of matter. But for some systems, even those that "look" like fluids such as, famously, oobleck – a mixture of corn starch and water – the simplest set of equations fail.
>
> Moreover, when one has a theory that successfully describes many systems, it is of great interest to find a similar system where the theory does not apply. In physics, such anomalies are invariably pointers to novel underlying phenomena and they help to stimulate scientific progress.
>
> For this reason, we felt that the anomalies that we saw with Gemini 2.5 Pro on vanilla addition were of great interest, and we do not view them as a negative indicator for our research program. This is the reason we emphasized them prominently in our paper! In our paper, we  made a conjecture for the underlying cause of the anomaly, and then gathered some evidence for this conjecture by performing additional experiments. It is of interest to understand these anomalies better. It is also of interest to find other tasks where the accuracy shows this erratic behavior.
>
> > All the tasks examined in this study share a similar algorithmic structure and use a small vocabulary, which may limit the model’s ability to generalize to other domains such as commonsense reasoning or factual question answering.
>
> The reviewer is right that our model applies to very simple tasks. But our reason for focusing on such tasks again traces back to the philosophy --- used with great success in physics --- of finding clean simple systems that are amenable to theoretical analysis. By studying these simple systems, we hope to discover some underlying principles that might eventually be applicable in more realistic settings. The fruitfulness of considering this simple collection of tasks is borne out by the fact that we have been able to develop a simple model that fits the error scaling curve extremely well.
>
> > The paper does not provide baseline models or alternative approaches for comparison with the proposed method.
>
> Thanks for highlighting this. A "baseline" error model is one where each output token has an independent probability of being erroneous.  We discussed this briefly in section 2.5. We will be happy to include more analysis, including a graph showing how our model fits the data much better than this baseline, in a revised version.
>
> **Response to questions**
>
> > What determines the parameter q? Can it be predicted based on the vocabulary structure or specific properties of the task?
>
> In our theory, the parameter $q$ is interpreted as the number of effective directions in which errors can be made. This parameter varies from task to task. In principle, given all the raw parameters of the model, it might be possible to predict this parameter but our approach is empirical; and we determine it experimentally. An analogy might be helpful: a fluid is described by a few empirical parameters such as its density and viscosity. In principle, given the chemical composition of the fluid and the fundamental intermolecular interactions, it is possible to predict these parameters.  But, from an “effective” perspective, what is remarkable is that all the intermolecular complexity reorganizes itself into a small number of relevant parameters that can be empirically determined.
>
> > If errors indeed accumulate through the attention mechanism, what types of architectural or training modifications might mitigate this issue?
>
> Our analysis suggests that a natural method of ameliorating errors might be to examine forms of "sparse attention". Such methods might reduce the accumulation of noise across long sequences in the attention mechanism. A second suggestion might be to try different positional embeddings that aid the model in accurately identifying the position of tokens in its context. We would be glad to expand the relevant discussion in our paper in a revised version.

---

> > ### Author Rebuttal · Reviewer_Ah8A · 2026-04-01
> >
> > The “physics-inspired” argument suggests that the theory is not yet applicable to realistic settings or larger vocabularies, and the failure cases observed in Gemini Pro indicate that the model’s assumptions may be overly rigid. Hence, I would prefer to keep my initial score.

---

> > > ### Author Response · Authors · 2026-04-06
> > >
> > > We would like to thank the reviewer for their comments. We agree with the reviewer that our analysis does not apply to  many “real world” tasks or for large vocabularies. We emphasized this in the first line of our abstract where we restricted the scope of our model to “tasks like arithmetic that require a deterministic output, and repetitive processing of tokens drawn from a small set of alternatives.”
> > >
> > > While the field is primarily focussed on improving LLMs for real-world tasks, we would like to take a step back and advance the theory of LLMs that currently lags the practical advances in the field. We found it satisfying that we could derive a simple two-parameter formula based on  a small set of theoretical assumptions that describes a system as complicated as an LLM, admittedly in a simple setting.
> > >
> > > We feel that in the initial stages of the theory, it is okay to step back from the most complex scenarios. In the real world, a leaf manifestly falls slower than a stone. Nevertheless, to advance the theory of projectile motion it is helpful to first imagine a setting without air resistance where they fall at the same rate, and then turn to real-world complications.
> > >
> > > Although a precise theoretical understanding of the error rate of LLMs on all sorts of real world tasks is currently out of reach,  we are hopeful that insights from simplified theoretical models of LLMs will eventually help in real-world applications.  We hope that the reviewer will encourage our efforts to advance the theory even if it is incipient at this stage.

---

### Official Review · Reviewer_N8QC · 2026-03-13

**Soundness:** 2
**Presentation:** 3
**Significance:** 3
**Originality:** 2
**Overall Recommendation:** 4
**Confidence:** 3

**Summary:**

The paper studies the relationship between complexity of a task and the resulting accuracy for LLMs. The authors use several repetitive tasks involving arithmetic, and scale complexity as a function of input length. They show that this relationship can be quantified with two parameters and fit a model with this. The error rate is framed as an accumulation of individual smaller errors over tokens. First, the authors define an effective model and compare it to an idealized model, and claim that the attention mechanism accounts for this accumulation of individual errors. Then, they show empirically that their model holds for a variety of tasks, conditioned on the LLM utilizing certain strategies for varying complexities of tasks.

**Compliance With Llm Reviewing Policy:**

Affirmed.

**Final Justification:**

After discussion, the authors addressed my concerns

**Key Questions For Authors:**

* What strategies were being employed by the Gemini Pro model for vanilla addition and the original multiplication setup?

* Why would errors not accumulate from a correct attention mechanism propagating incorrectly computed MLP outputs at each token that each have a slight error?

* What is the relationship between the number of tokens generated in each setting and with the input length complexity?

**Limitations:**

yes

**Strengths And Weaknesses:**

The paper does successfully show that accuracy can be represented as a function of complexity and two other parameters in their settings. The empirical results do show the same relationship occurring across the three models and across tasks. It is unclear how the attention mechanism specifically can be attributed to this issue though, as opposed to other parts of the transformer (embeddings, MLPs). Errors in the non-linearities of the MLP for example could propagate between tokens despite precise attention, and lead to similar effects. Using input length as a measure of complexity while ignoring the number of generated tokens is an oversimplification as well: certain operations can be done within a forward pass, while others may require several. The results also seem to require the model following the same strategy regardless of complexity, excluding certain classes of errors mentioned in the introduction like pattern matching. The improvement in model accuracy is due to prompting with a certain proven strategy, which results in a limited takeaway of how to use this model for improvements.

---

> ### Author Rebuttal · Authors · 2026-03-30
>
> We would like to thank the reviewer for their comments.
>
>
> > unclear how the attention mechanism specifically can be attributed to this issue though, as opposed to other parts of the transformer (embeddings, MLPs).
> > Why would errors not accumulate from a correct attention mechanism propagating incorrectly computed MLP outputs at each token that each have a slight error?
>
> We specifically address this issue from an intuitive perspective in the main text below Assumption A4 and then address it quantitatively in the Appendix.  Equation 22 in the Appendix displays the effect of errors in the feedforward layer. The scaling of the variance with c displayed in Assumption A4 arises from the attention mechanism, and is not altered by the feedforward layer. If the reviewer has a specific objection to any of the relevant equations, we will be happy to address them.
>
> > Using input length as a measure of complexity while ignoring the number of generated tokens is an oversimplification as well
> > What is the relationship between the number of tokens generated in each setting and with the input length complexity?
>
> We respectfully disagree with this interpretation of our analysis that we take the input length as the complexity parameter and ignore the number of generated tokens.
>
> Our complexity parameter is defined in Section 2.3 as “the minimal number of tokens that must be processed by the idealized model, including CoT tokens.” The idealized model is different from the actual LLM and is defined in Section 2.2. Its existence is guaranteed by the references there. In the tasks at hand, this minimal number of tokens is expected to scale linearly with $\ell_\text{in}$ or $\ell_\text{out}$. We exploit the reparametrization invariance of our formula (2nd para in Section 2.3) under linear rescaling of $c$ to then use either $\ell_\text{in}$ or $\ell_\text{out}$ as the complexity parameter. Please note that the definition of $\ell_\text{in}$ itself involves an abstraction since it is the “effective” part of the input, as defined below equation (3) and not the total prompt length.
>
> To be clear: the actual number of thinking tokens produced by the LLMs is not necessarily a good measure of complexity since those tokens depend on a variety of factors and perform functions including error corrections. In our initial analysis, we indeed considered the number of total generated tokens of the actual LLM (not of the idealized model) as a possible complexity parameter, but did not find any clear scaling behaviour of the error rate with this, which then motivated us to consider the $\ell_\text{in}$ and $\ell_\text{out}$ of the idealized model as a more reliable complexity parameter. We would be happy to add a graph discussing this in a revised version of the paper. We focus on this theoretical minimum number of tokens in line with our perspective that it is productive to abstract away from the enormously complex functioning of the LLM to a simple effective model that captures its behavior in specific settings.
>
> This is precisely the perspective adopted in physics: rather than focusing on all the complex raw parameters of a system we seek to find a lower-dimensional effective description that accurately models the system in specific experiments.
>
>
>
>
> > What strategies were being employed by the Gemini Pro model for vanilla addition and the original multiplication setup?
>
> Our methodology does not allow us to determine the internal algorithm used by LLMs (we are unaware of any methodology that can answer this question as well). Our paper is a study of LLMs from a natural science perspective where we seek to quantitatively describe its error rate on a very simple class of problems. The fact that there is no apparent scaling behavior for the error rate as we increase the complexity in this particular task (Figure 5(c)) led us to the suggestion that Gemini Pro might be using inconsistent internal algorithms, and we performed a followup experiment with a different prompt --- which instructed the model to utilize a specific algorithm --- to check this hypothesis. With this new prompt, we observe a scaling behavior (Figure 6(c)) providing evidence supporting our claim. Beyond this, our methodology does not allow us to determine the internal algorithm used by LLMs. The precise prompts given to Gemini Pro and other models are available in Appendix D.

---

> > ### Author Rebuttal · Reviewer_N8QC · 2026-04-05
> >
> > 1.) Thank you for this pointer to the appendix, this is helpful
> >
> > 2.) I found this very clarifying and thank the authors for this response. I'm convinced my interpretation was somewhat incorrect/not generous.
> >
> > Thank you to the authors for their response; my questions are resolved and I'm more positive about the paper. I've updated my score accordingly

---

> > > ### Author Response · Authors · 2026-04-06
> > >
> > > We are very glad that the reviewer found our reply helpful and does not have any remaining concerns. We would also like to thank the reviewer for the revised score although we are currently unable to view the revised score on the portal.

---

### Decision · Program_Chairs · 2026-04-30

**Decision:**

Accept (regular)

**Comment:**

This paper investigates errors in transformer LLMs. There was consensus that the work offers an interesting model of such errors, with compelling empirical results. The work is somewhat limited in the scope of tasks considered, and some reviewers did not find the assumptions used in the theoretical account plausible. Major concerns were sufficiently addressed by the authors in the rebuttal, and despite its limitations this work offers an interesting view on a practically important phenomenon.